https://doi.org/10.1038/s42003-020-01284-7　**OPEN**

# Phylogenetic inference enables reconstruction of a long-overlooked outbreak of almond leaf scorch disease (*Xylella fastidiosa*) in Europe

Eduardo Moralejo [1✉], Margarita Gomila [2], Marina Montesinos[1], David Borràs[3], Aura Pascual[1], Alicia Nieto[3], Francesc Adrover[3], Pere A. Gost[4], Guillem Seguí[2], Antonio Busquets[2], José A. Jurado-Rivera [5], Bàrbara Quetglas[4], Juan de Dios García[4], Omar Beidas[4], Andreu Juan[4], María P. Velasco-Amo[6], Blanca B. Landa [6] & Diego Olmo[3]

The recent introductions of the bacterium *Xylella fastidiosa* (*Xf*) into Europe are linked to the international plant trade. However, both how and when these entries occurred remains poorly understood. Here, we show how almond scorch leaf disease, which affects ~79% of almond trees in Majorca (Spain) and was previously attributed to fungal pathogens, was in fact triggered by the introduction of *Xf* around 1993 and subsequently spread to grapevines (Pierce's disease). We reconstructed the progression of almond leaf scorch disease by using broad phylogenetic evidence supported by epidemiological data. Bayesian phylogenetic inference predicted that both *Xf* subspecies found in Majorca, *fastidiosa* ST1 (95% highest posterior density, HPD: 1990–1997) and *multiplex* ST81 (95% HPD: 1991–1998), shared their most recent common ancestors with Californian *Xf* populations associated with almonds and grapevines. Consistent with this chronology, *Xf*-DNA infections were identified in tree rings dating to 1998. Our findings uncover a previously unknown scenario in Europe and reveal how Pierce's disease reached the continent.

[1] Tragsa, Empresa de Transformación Agraria, Delegación de Baleares, 07005 Palma de Majorca, Spain. [2] Microbiology (Biology Department), University of the Balearic Islands, 07122 Palma de Majorca, Spain. [3] Serveis de Millora Agrària i Pesquera, Govern de les illes Balears, 07009 Palma de Majorca, Spain. [4] Servei d'Agricultura, Conselleria d'Agricultura, Pesca i Alimentació; Govern de les illes Balears, 07006 Palma de Majorca, Spain. [5] Laboratory of Genetics (Biology Department), University of the Balearic Islands, 07122 Palma de Majorca, Spain. [6] Institute for Sustainable Agriculture, Consejo Superior de Investigaciones Científicas (IAS-CSIC), 14004 Córdoba, Spain. ✉email: emoralejor@gmail.com

Almond trees, an important icon of the agricultural landscape of Majorca, Balearic Islands (Spain), have experienced severe decline and mortality over the last 15 years[1]. The almond disease was preliminarily studied from 2008 to 2010 in the main initial focus in Son Carrió, east of Majorca[1]. Although no etiological agent was ascribed, this disease was associated with a complex of fungal trunk pathogens[2,3] and their interactions with known disease-predisposing factors such as prolonged drought and tree aging[4,5]. This early disease diagnosis, however, was recently challenged after the detection of *Xylella fastidiosa* (*Xf*) in Majorca in October 2016[6]. More than 119 almond samples tested positive for *Xf* in a 2017 analysis conducted by the Balearic Islands Official Plant Health Laboratory (LOSVIB), which raised suspicion that *Xf* could underlie the sudden emergence of this unprecedented disease[7].

*Xf* is a xylem-inhabiting bacterium that is transmitted exclusively in nature by xylem-fluid-feeding insects[8] and by budding and grafting[9–11] and experimentally by wound inoculation. *Xf* is a genetically diverse species made up of three subspecies[12–14] and can potentially infect more than 500 plant species[15]. Furthermore, each subspecies is formed by multiple genetic lineages, grouped as sequence types (ST), each with different host ranges, although most of them infect one or several known hosts[16,17]. Before 2013, *Xf* was officially known to cause economically important diseases on crops, ornamental and landscape plants only in the Americas[18] and Taiwan[19]. In 2013, the report of a lethal outbreak affecting olive trees in Apulia, Italy, raised alarms about the threat posed by *Xf* to Mediterranean agriculture[20–22]. A new disease named 'olive quick decline syndrome' is caused by a strain of the subspecies *pauca*, mainly transmitted by the insect *Philaenus spumarius* (Hemiptera: Aphrophoridae)[20,21,23]. Since 2013, various *Xf* genotypes belonging to *fastidiosa*, *pauca*, *multiplex* and 'sandyi' subspecies have been intercepted in and/or introduced into Europe[24–29].

Following the EU mandatory annual surveys (Decision EU 2015/789), more than 7287 plant samples comprising 274 plant species have been analysed for *Xf* at the LOSVIB from 2016 to 2019. As a result, three *Xf* subspecies have been identified: (i) *Xf* subsp. *fastidiosa* ST1, which causes Pierce's disease on grapevines and almond leaf scorch disease (ALSD) in California[30] and has been recorded only on Majorca Island; (ii) a new ST of *Xf* subsp. *pauca* (ST80) that is found only on Ibiza Island, mainly on olive trees (*Olea europaea*); and (iii) another novel ST of *Xf* subsp. *multiplex* (ST81) that is closely related to ST6 and is present in the Majorca and Menorca islands[31]. In addition, there have been two single detections of *Xf* subsp. *multiplex* ST7 in Majorca. Today, 23 hosts, including almond (*Prunus dulcis*), grapevine (*Vitis vinifera*), fig tree (*Ficus carica*) and wild and cultivated olive tree (*Olea europaea* subsp. *sylvestris* and subsp. *europaea*, respectively), have been found to be infected by different subspecies in the Balearic Islands[32].

ALSD was first reported in California in the mid-1930's[9,30], and more recently there have been confirmed outbreaks in Iran[11], in Alicante in eastern Spain[29] and in the Hula Valley, Israel[33]. In California, ALSD is transmitted by xylem specialist insect species commonly known as sharpshooter leafhoppers (Hemiptera: Cicadellidae) and by the meadow spittlebug, *P. spumarius*, although the latter seems to play a secondary role in disease transmission[34]. Infected trees initially show leaf scorch symptoms and experience a progressive decline with frequent death over the following 3–8 years[9]. In addition to ST1, ALSD is also caused by strains of the subsp. *multiplex* belonging to ST6 and ST7, which form a clonal complex integrated into a monophyletic clade[31,35]. The phylogeny and likely origin of these strains have been reviewed by Nunney et al.[36,37], and the complete genomes of several representative ALSD strains have been sequenced[29,38,39].

ST1 strains infect grapevines and can cross-infect almonds as well, whereas almond strains of ST6/7 do not cause disease on grapevines[30]. Both subspecies, however, coexist sympatrically in almond orchards in the San Joaquin Valley of California[40,41].

To understand the timescale of *Xf* epidemics in Majorca, the spatiotemporal relationship between the disease attributed to fungal trunk pathogens and ALSD needs to be addressed. In this study, we investigated whether the almond disease was actually due to the introduction of *Xf* —i.e. the spreading pathogen hypothesis—with its subsequent pathogen-induced drought by occlusion of the xylem vessels or instead caused by 'endemic' fungal pathogens activated by abnormal environmental changes. Preliminary observations of images from Google street view between 2012 and 2017 strongly suggest that ALSD symptoms precede shoot and branch diebacks, which were previously attributed to fungi. We therefore hypothesized that fungal trunk infections would likely be implicated in the advanced stage of ALSD. If true, this hypothesis would necessarily imply the emergence of almond decline in Son Carrió as an upper boundary date for *Xf* arrival. According to a regional plant health advisor (T. Melis, *personal communication*), the first abnormal clusters of dying almond trees were noted by around 2003 in Son Carrió. In support of the spreading pathogen hypothesis, we localized two orchards in which Californian almond varieties were grafted onto local almond rootstock in the 1990s, including one orchard in Son Carrió. Moreover, in July 2017, we determined that the subspecies *fastidiosa* ST1 and *multiplex* ST81 and ST7 caused ALSD in Majorca. Interestingly, coinfections by two of those STs have been reported only in California[40]. Two recent reports on the genomes of three isolates of *Xf* subsp. *fastidiosa* ST1, collected from Majorca, reported the 38 kb plasmid pXFAS_5235, with the highest sequence similarities to the conjugative plasmid pXFAS01 from isolate M23 causing ALSD in California[39,42]. In addition, a close genetic relationship between Majorcan *Xf* isolates belonging to ST1 and ST81 and Californian *Xf* isolates related to Pierce's disease and ALSD was confirmed recently by phylogenetic analysis based on core genomes[6,31].

In this work, we followed *Xf* infections in Majorca, retracing almond decline over time. Our first aim was to determine whether *Xf* was actually the causal agent of almond decline. We investigated the epidemiology of ALSD and its relationship with fungal trunk pathogens. Our second objective was to shed light on the cryptic stage of *Xf* establishment on the island. We tracked the *Xf*-DNA harboured inside the growth rings of infected trees through dendrochronological analysis combined with *Xf*-specific quantitative polymerase chain reaction (qPCR) assays. The presence of *Xf*-DNA in dated rings was used as an infection marker to calculate the survival time of almond trees and to calibrate the approximate timescale of the ALSD in Majorca. Finally, we aimed to estimate the introduction date of both subspecies into Majorca using time-calibrated phylogenies. The whole-genome sequences of 22 *Xf* isolates recovered from almond trees with ALSD symptoms and from other hosts on the islands of Majorca and Menorca were compared with the available genomes of strains of the subspecies *fastidiosa* and *multiplex* causing ALSD and Pierce's disease in California. We show how one of the oldest established unreported *Xf* outbreaks in Europe could have started after infected buds of Californian varieties were grafted onto local rootstocks around 1993. From this introduction event, the meadow spittlebug, *P. spumarius*, presumably transmitted *Xf* to multiple hosts, including grapevines, wild olive and fig trees.

## Results and discussion
**ALSD is widespread in Majorca.** The summer of 2017, following the *Xf* official detection in October 2016, provided a good

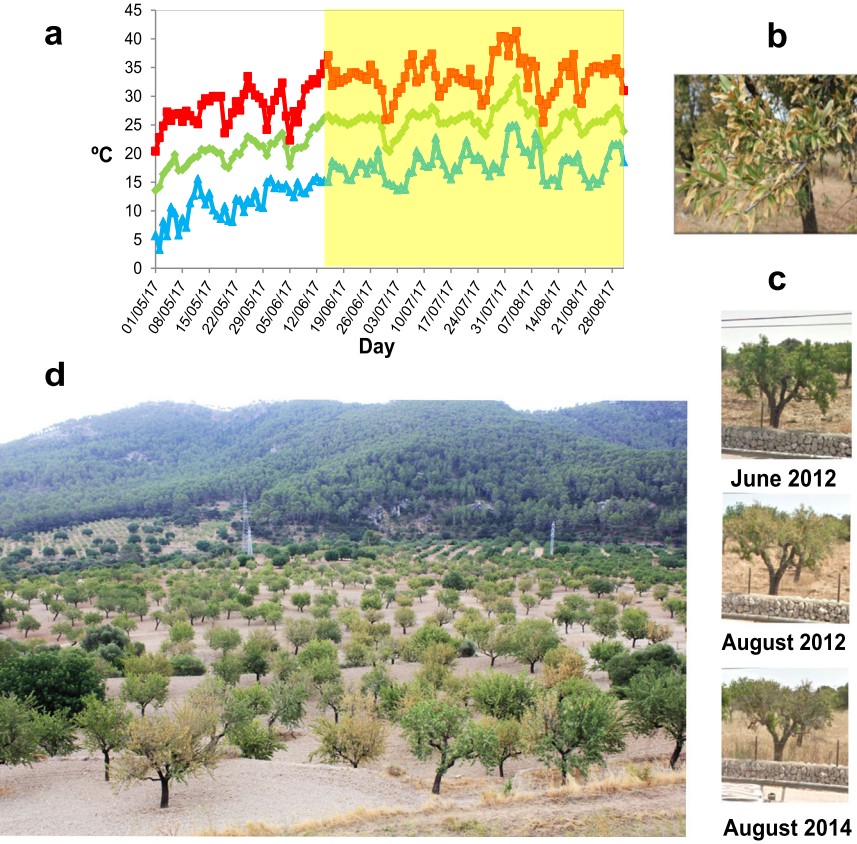

**Fig. 1 Characterization of almond leaf scorch disease (ALSD) in Majorca caused by *Xylella fastidiosa*. a** Maximum (red line), mean (green line) and minimum (blue line) temperatures in Majorca triggering the emergence of scorch symptoms in June 2017. **b** Detail of an almond branch with leaf scorch symptoms. **c** Image sequence from systemic scorch symptoms in June 2012 to severe dieback in August 2012 and tree death in 2014 obtained from Google street view. **d** Image taken in July 2017 at Puigpunyent, ~70 km from the main focus at Son Carrió. The orchard shows infected trees with systemic symptoms of ALSD and 'golden death' with few diebacks.

scenario for testing whether *Xf* was implicated in the almond decline. An episode of high water demand (three consecutive days of $T_{max} > 33\,^{\circ}C$ and $R_H < 60\%$; Fig. 1a) in mid-June 2017 triggered the synchronous onset of ALSD symptoms throughout Majorca. Scorch affected both the leaves attached to single terminal shoots and, more frequently, those on main branches or on the whole crown (Fig. 1b, c). In mid-July, distinctive leaf scorch and 'golden death' symptoms, as described elsewhere[9], were prevalent (Fig. 1d). Such an explosion of ALSD symptoms concurred with a European Commission audit, which motivated changes in the recommendation from eradication to containment measures [DG (SANTE) 2017–6216; EU 2017/2352]. ALSD symptoms emerged two weeks later in the summer of 2018 than in 2017 due to wetter and milder late-spring early-summer weather, stressing the annual variability in ALSD symptom expression and other *Xf*-related diseases driven by environmental factors[43,44].

**ALSD is caused by two *Xf* subspecies in Majorca.** Seventy-six out of 105 almond leaf samples across Majorca brought to the LOSVIB in the summer of 2017 tested positive for *Xf* in qPCR analyses (using both Harper et al.[45] and Francis et al.[46] qPCR protocols). Multilocus sequence typing (MLST) analysis ($n = 23$) based on DNA extracts from almond leaf samples and some recovered *Xf* isolates revealed three STs: ST1 belonging to subsp. *fastidiosa*; a new ST, ST81, belonging to subsp. *multiplex*; and ST7 belonging to subsp. *multiplex*, detected only once (https://figshare.com/s/0b571280bcd7738a524d; 10.6084/m9.figshare.12378302). ST81 differs from known Californian isolates within the 'almond' clonal complex ST6 and ST7 in only one of the seven alleles used for MLST. ST81 shared an identical MLST profile with the isolate 'Fillmore' (Acc. n°: CP052855.1) recovered from an olive tree in California. Both ST1 and ST81 were well distributed across the island, with ST81 also infecting the widespread wild olive trees among other hosts, whereas strain ST1 caused Pierce's disease as well[6]. No clear spatial pattern in their distribution was observed, suggesting long-term spread and mixed-infection scenarios (Supplementary Fig. 1).

In total, 55 isolates (62%; $n = 89$ attempts) were obtained from different almond trees recovered from petioles of symptomatic leaves (qPCR positive) in the summers of 2017 and 2018. Colony morphotypes similar to those reported by Chen et al.[40] in almond orchards of California were distinguished on periwinkle wilt GelRite (PWG) medium, i.e. almond A-genotypes and grape G-genotypes corresponding to the subsp. *multiplex* and *fastidiosa*, respectively (Supplementary Fig. 2). Both subspecies were once isolated from the same tree, and in the same orchard; *Xf*-DNA of both subspecies was detected in the growth rings of two out of three trees examined, strongly indicating frequent coinfections (Supplementary Fig. 2).

**The current ALSD incidence and tree mortality preclude a recent introduction.** By counting the trees that showed disease symptoms previously attributed to fungal trunk pathogens as an advanced stage of ALSD (see Methods), the incidence for the disease in 2017 was visually determined to be approximately 79.5% ± 2.0 (mean ± SEM) (Supplementary Data 1; Supplementary Software 1). From this percentage, we estimated that ~1,250,308 almond trees, including dead trees, would have been

infected by *Xf* (2017 almond plantation census: 19,417 ha with an average density of 81 trees ha$^{-1}$; https://www.mapama.gob.es). Similarly, we extrapolated that at least ~552,869 dead trees remained in the fields (Supplementary Data 1). This number is actually an underestimation of the trees that have died since the beginning of the outbreak, given that many dead trees were removed from the field. Disease incidence ranged between 16.6 and 100%, with half of the orchards exhibiting ALSD incidences over 90 and 30% mortality (Supplementary Data 1). ALSD incidence affected trees of different ages equally ($P = 0.50$); however, younger plantations (trees ≤ 30 years) suffered less mortality than plantations with trees older than 30 years ($\chi = 5.37$; df = 1, $P < 0.020$).

Such a high ALSD incidence across the island suggest a relatively old entry of the pathogen. To estimate the rate of disease spread, we needed some historical reference to the ALSD incidence. The Google street view panoramic-image repository has been used to assess the distribution and prevalence of other pest and diseases in a territory such the pine processionary moth in France[47]. This approach provided the means to approximately assess the ALSD incidence and mortality in 2012, given that the pictures covered a large part of the territory that year (see Methods). Thus, 249 orchards distributed throughout the island were visually examined, and the average incidence of ALSD was estimated at 53.4% ± 1.6 (mean ± SEM) for 2012 (Supplementary Data 2). This figure already exceeded the inflection point in the disease progression curve projected in a logistic model (y) = ln (y/ (1 − y)), where y is the proportion of infected trees in a predominant tree-to-tree transmission, while the rate of new infections decreases in proportion to the number of remaining uninfected almonds.

We predicted a greater accumulation of *Xf*-infected plants in those areas of previous almond decline. As expected, the orchards closest to the putative main focus area (radius < 20 km) and therefore with a longer exposure time to *Xf* showed a higher incidence ($\chi = 25.21$, df = 1, $P = 0.0001$; Supplementary Data 1) and mortality ($\chi = 26.49$, df = 1, $P = 0.0001$; Supplementary Data 1), supporting the hypothesis that Son Carrió was an expanding focus. Likewise, the incidence and severity of Pierce's disease had previously been shown to be higher in orchards closer to the main propagation centre in Son Carrió[6]. To further capture the spatial-temporal spread of ALSD, we mapped the estimated disease incidence (%) and mortality (%) distribution among orchards for 2012 and 2017 using QGIS[48] geographic information system software (Fig. 2). A careful review of the maps revealed a heterogeneous dispersion with several emerging expansion centres, in addition to the main centre in the Son Carrió area (Fig. 2). This distribution pattern could be explained considering two factors: (i) the occasional long-distance dispersal through infected grafts—a common practice in Majorca—at the beginning of the epidemic and (ii) the regional differences in the rate of expression of disease severity and mortality due to environmental and biological factors, such as soil type, agricultural practices, differing susceptibilities of almond varieties and precipitation. In the case of Son Carrió, the combined effect of an early introduction, low rainfall, and shallow soils could have increased the severity of ALSD and therefore resulted in an earlier disease awareness than in any other area of the island.

**ALSD precedes symptoms caused by fungal trunk pathogens.** We initially doubted whether the trunk pathogenic fungal complex could be the cause of the almond epidemic, and *Xf* infections were an additional factor that would aggravate the disease. In mid-summer 2017, we noted that trees with ALSD symptoms commonly intermingled in the same orchard with others that exhibited different stages of general decline, shoot and branch diebacks, or a combination of both (Supplemental Data 1 and 3). In our evaluations of disease incidence, we found a significant statistical dependence between ALSD incidence (%) and tree mortality (%) within orchards (2017 disease incidence assessment: Spearman's rank correlation coefficient: ρ (126) = 0.88; $P < 0.0001$; 2012 disease incidence assessment ρ (249) = 0.89; $P < 0.0001$; Supplementary Data 1 and 2). Friedman's test showed that there were significant differences among repeated measures of tree severity scores in the time series, which corresponded to the sequences of symptom development, from leaf scorch to shoot and branch diebacks to tree death, over time ($\chi = 42.41$, df = 4, $P < 0.0001$: Kendall's coefficient of agreement = 0.69; Supplementary Fig. 3; Supplementary Data 3). Furthermore, ALSD symptoms preceded shoot and branch death in 96% of cases, while trees without ALSD symptoms remained healthy until images were no longer available (Supplementary Data 3). Together, these results suggest a continuum in the process from systemic *Xf* infection to tree death, as illustrated in Fig. 1c. Although these general observations do not prove causality, they show that *Xf* infections mainly preceded any fungal trunk symptoms, thus corroborating the explanation for *Xf*-induced water stress as a disturbance factor favouring fungal activation from an endophytic commensal lifestyle to a virulent pathogenic stage[4].

**A combination of dendrochronology and qPCR revealed *Xf* old infections.** Some direct evidence of *Xf* spread before 2003 emerged after examining the presence of *Xf*-DNA in the tree rings of the wood sections (see Methods). We plotted for each felled tree (n = 34) the relationship between the quantitative cycle (Cq) values obtained in the qPCR and the dating of the tree rings from which sawdust samples were taken. Consistent with the assumption of annual cyclic colonization by *Xf* of newly formed xylem tissue in spring (see Methods), the *Xf*-DNA concentration measured in Cq values significantly increased from the oldest to the youngest rings (linear model: $F_{1,183} = 66.11$, $P < 0.0001$; Supplementary Figs. 4 and 5; Supplementary Data 4A, B). For example, *Xf*-DNA extracted from growth rings corresponding to 2008 ± 2 was amplified by qPCR in 12 of the 34 trees examined, while this number decreased to nine trees in the 2004 ± 2 rings. In a tree sample (XYL 739/17), *Xf*-DNA was amplified (Cq = 29) from a growth ring dated 1998 ± 2. In contrast, *Xf*-DNA was not detected in rings dated prior to 1995 (Supplementary Data 4A, B). Additionally, specific primers and probes targeting genome-specific regions in a duplex qPCR assay[45] allowed the differentiation between *fastidiosa* and *multiplex* subspecies in rings of 25 trees, with nine and 19 trees showing infection by subspecies *fastidiosa* and *multiplex*, respectively, and three trees showing a mixed infection. The analysis enabled dating the infection back to 1998 (subsp. *fastidiosa* ST1) and before 2000 (subsp. *multiplex* ST81) in Maria de la Salut and Binissalem, respectively (Supplementary Data 4C). After plotting the infection frequencies of dated rings in the 34-almond samples, we observed that the infection frequency sharply increased from the inner older rings to the outer rings, fitting this distribution to a logistic disease progression curve from 1993 to 2017 (Fig. 3a; Supplementary Data 4A). Notably, 1993 was the extrapolated date for epidemic onset (disease incidence < 0.01%) in the logistic model and very closely fits the date of the most plausible introduction event (Fig. 3b).

**Survival analysis.** Taking advantage of the *Xf* qPCR data from dated trunk rings for each felled tree, we calculated the time from infection to tree death (decrepitude). When the trees were cut down in 2017 and 2018, most were still alive, so they were right censored in the survival analysis (Supplementary Data 4D).

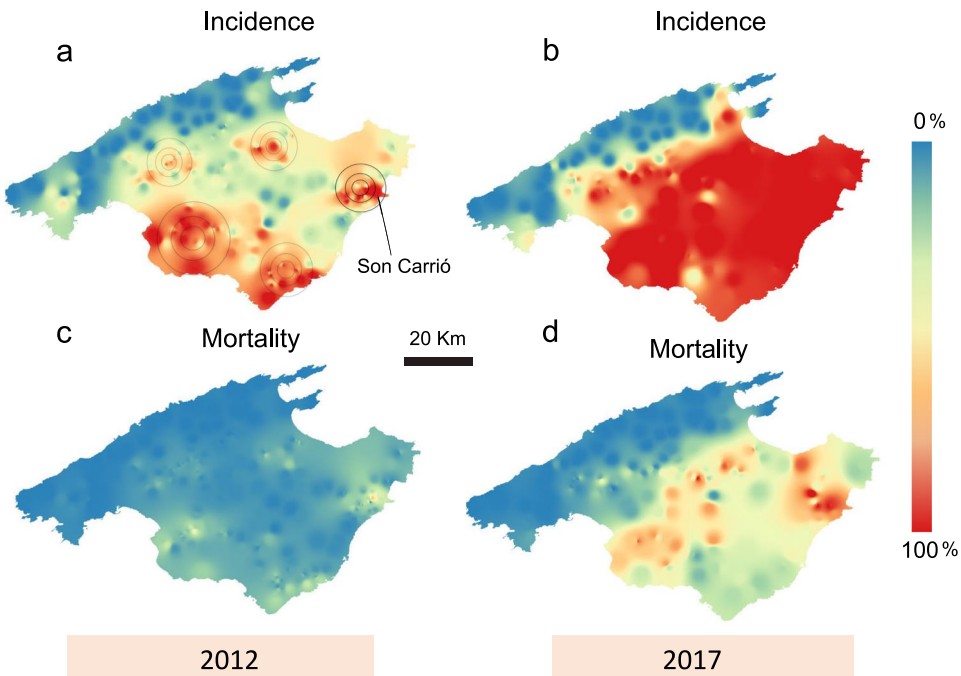

**Fig. 2 Inverse-distance-weighting interpolation map representing the spatial distribution for almond leaf scorch disease (ALSD) incidence and almond tree mortality within orchards across Majorca in 2012 and 2017.** Data were collected from observations ($n = 249$ independent orchards) on Google street view in 2012 (**a, c**) and from direct field observations ($n = 126$ independent orchards) in the summer of 2017 (Supplementary Data 1 and 2). **a** In 2012, several spreading foci were observed with high disease incidence. **b** ALSD incidence is widespread across the island in 2017, showing a gradient from east to west. **c** Red hotspot of tree mortality in Son Carrió in 2012, where the almond decline was first detected around 2003. **d** Large areas showing tree mortality mostly in the east and south of the island.

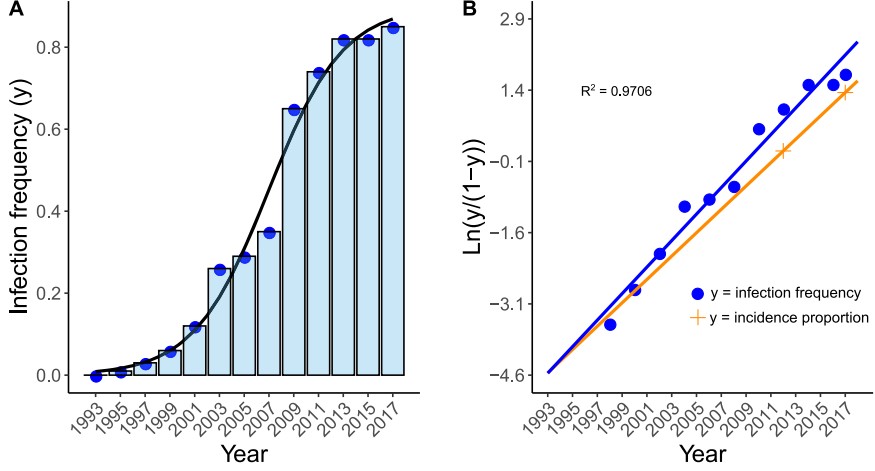

**Fig. 3 Estimation of almond leaf scorch disease progress curve in Majorca.** Estimation inferred from the proportion of tree rings infected by *Xylella fastidiosa* (*Xf*) at 2-year intervals in 34 sampled almond trees (Supplementary Data 4). **a** The arrangement of infection proportions (bars) fits the disease progress curve expected in a logistic model. **b** Infection proportion (y) was log transformed and plotted against the year of the tree rings (blue dots). Disease incidence estimations for 2017 and 2012 were included (orange dots and line). The intersection between the regression line and x-axis at −4,6 (=0.01 disease incidence) provides a rough estimate of the likely time of *Xf* introduction.

The Kaplan–Meier median (50%) survival estimate was 14 years (95% CI: 11–17), considerably longer than the 3–8-year-period reported by Mircertich et al.[9] in California (Fig. 4). Our results nevertheless were closer to those reported by Sisterson et al.[43] in a 6–7 year monitoring of almond plantations affected by ALSD in California, where they found that 91% of infected trees survived to the end of the study.

**Both *Xf* subspecies are pathogenic and reproduce ALSD symptoms.** To explain the high disease incidence and mortality

observed, the 'spreading alien pathogen'" hypothesis requires *Xf* strains to be broadly pathogenic to more than 87 local and 23 nonlocal almond varieties that are growing throughout the island and identified in the almond germplasm collections in Majorca. The extent of their pathogenesis, however, was only partly revealed in the inoculation assays of 2018. Strain XYL 2055/17 (ST1) from grapevine caused typical almond leaf scorch symptoms in 24 out of the 160 saplings in nine out of 11 varieties 16 weeks after inoculation (Supplementary Table 1). Two out of eight plants of *cv.* Filau inoculated with strain XYL1752/17

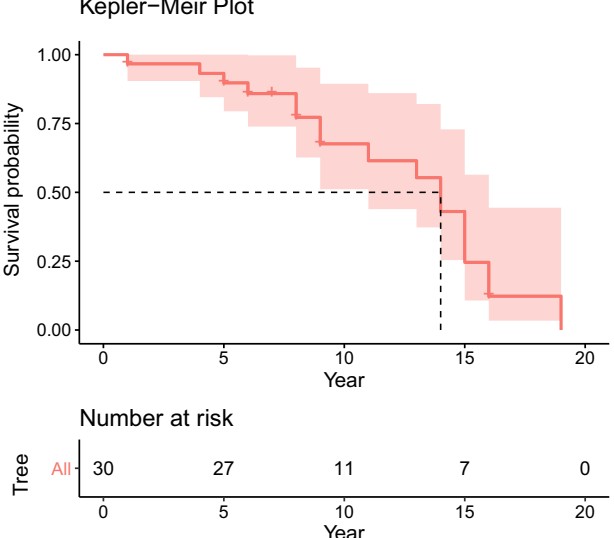

**Fig. 4 Survival analysis of almond trees infected by *Xylella fastidiosa* in Majorca.** The time of infection for each tree ($n = 30$) was determined by qPCR assay on DNA extract from sawdust obtained from dated almond growth rings. Most of the trees were symptomatic and alive at felling and thus right censored.

belonging to ST81 in October 2017 developed scorch symptoms in August 2018. In both inoculation experiments, the pathogen could be detected by qPCR on leaves more than 10 cm from the point of inoculation and reisolated from the petioles, thus fulfilling Koch's postulates. In 2019, we realized that the almond varieties used in the inoculation experiments were among the less-susceptible varieties in the field after monitoring ALSD infection incidence in almond germplasm collections distributed in four fields around Majorca, which may explain the low rate of disease symptoms observed. Seventy-eight of 110 almond varieties naturally exposed to *Xf* for several years showed ALSD symptoms, and infections by the bacterium were confirmed by qPCR (Supplementary Table 2).

**ALSD is transmitted by spittlebugs**. Based on 'olive quick decline syndrome' epidemiological studies in Italy[21] and Pierce's disease in Majorca[6], *P. spumarius* was the main candidate vector for ALSD. In the transmission experiment, *P. spumarius* adults reared from nymphs and caged on ALSD symptomatic trees vectored *Xf* to healthy almond saplings. *Xf* was detected by qPCR in seven out of eight almond plants exposed to 'infected' *P. spumarius* and reisolated from symptomatic leaves on six of those plants. Both control plants exposed to uninfected insects ($n = 2$) and almond plants unexposed to insects ($n = 2$) were qPCR negative. *Xf* DNA was detected in all *post-mortem P. spumarius* adults ($n = 35$) used in the inoculation treatments, while all insects ($n = 10$) in two control replicates proved PCR negative. On the other hand, we investigated the prevalence of *Xf* subspecies in infected *P. spumarius* adults in the field by analysing the MLST of *Xf*-DNA within a subsample of 55 adults that were *Xf* positive for qPCR collected in or near three almond orchards. Among the positive spittlebugs analysed by MLST (25 samples), for 19 of them, some MLST loci could be amplified; two were infected by subsp. *multiplex* with alleles belonging to ST81, and 17 insect samples had alleles of subsp. *fastidiosa* ST1 (https://figshare.com/s/0b571280bcd7738a524d; 10.6084/m9.figshare.12378302).

**Fungal trunk pathogens are nonspecifically related to almond decline**. In previous works[2,3,49], Koch's postulates were fulfilled for most of the pathogenic trunk fungi found associated with almond decline; however, none of them could be ascribed as the aetiological agent of the decline. Therefore, the term 'complex disease' was used in those studies. We collected data from the work of Olmo et al.[49] to determine whether the regional diversity of fungal species associated with almond decline could be related to *Xf* infection. We observed that fungal diversity increased in those areas where ALSD severity was the highest and therefore with the longest exposure time to the pathogen (Fig. 2; Supplementary Fig. 6). Both of these observations agree with the explanation that the endophytic fungi would be activated to their pathogenic phase as a result of the stress induced by the *Xf* infection in the tree (i.e. the disturbance factor). Related to this lack of specificity of wood fungal pathogens, we found that two of the most pathogenic and frequently isolated fungal species on almonds in Majorca, *Neofusicoccum parvum* and *Pleurostomorpha richardsiae* (Supplementary Table 3), were also initially implicated in 'olive quick decline syndrome' caused by *Xf* in Apulia, Italy[50]. In contrast, in the nearby province of Foggia, Apulia (300 km from Gallipoli), still free of *Xf*, the same fungi provoked olive decline without causing death[50].

**Biologically induced water stress versus climatic drought**. Among farmers, the almond decline was generally attributed to the increase in drought periods due to climate change. Drought has also been partly associated with the appearance of the almond wood fungal complex[49]. If there was a relationship between drought and almond decline, an increase in drought episodes would be expected beginning at the time the disease was first noticed around 2003 (see Methods; Supplementary Data 5). To test this, we compared the crop average cumulative water deficit index (CWDi) during the periods 1988–2002 (before decline) and 2003–2017 (after decline). We found that the mean CWDi before 2003 was lower (CWDi = −263.4), indicating greater water stress, than that after 2003 (CWDi = −235.7), but this difference was not statistically significant ($t = 1.25$, df = 28, $P = 0.22$). Because it could be argued that drought periods prior to 2003 could have exceeded an irreversible threshold that activates pathogenic fungi to the present day, we looked for similar drought periods before 1988. We found that other severe episodes of water deficit anomalies occurred during 1963–1968 and 1981–1984[5], without triggering any almond decline in the following decade. Surprisingly, the wettest period (i.e. highest CWDi) of the general data series between 1988 and 2017 occurred between 2004 and 2010 (see Supplementary Data 5), matching the period of the exponential growth of the ALSD outbreak (Fig. 3a). Regardless of the effect of weather as the main driver of ALSD epidemics, the most plausible explanation for the emergence of fungal trunk pathogens is the disturbance (i.e. induced drought) caused by infection and colonization of xylem vessels by *Xf*.

**From the 1990s to 2003: the cryptic period**. Taken together, the data presented thus far strongly support the "spreading alien pathogen" hypothesis (i.e. *Xf* introduction) as the original cause of almond decline in Majorca (beginning in 2003[1]) over previous explanations of 'endemic' fungal trunk pathogens being activated by abnormal environmental changes. This almond disease showed a well-defined main focus in Son Carrió, several spreading centres with wave-pattern fronts, and a clonal genetic population structure, all of which are distinctive traits for an "introduced spreading pathogen". We conclude that ALSD was overlooked and confounded with the advanced stages of *Xf* infection when the action of a complex of fungi aggravated the

disease. As occurred with the initial diagnosis of 'olive quick decline syndrome' in Apulia, Italy[50], the almond decline was also associated with a complex of fungal trunk pathogens[1,2], comprising a similar assortment of fungal species in both diseases (Supplementary Table 3). This is a key piece in the reconstruction of the ALSD epidemic and the main chronological link between the introduction event and the awareness of the presence of *Xf* in Majorca in October 2016.

Similar to other introduced plant pathogens, *Xf* underwent a lag period of inoculum build-up at the early epidemic stage before damage became apparent[51]. Given our median survival estimate of 14 years for ALSD-infected trees, the disease emergence in Son Carrió around 2003, the lack of detection of *Xf*-DNA in growth rings prior to 1995 and the disease logistic model projecting 1993 as the time with a 0.01 disease incidence (Fig. 3b), we proposed the beginning of the 1990s as the most likely time for *Xf* introduction. In 2017, we performed an epidemiological investigation to find connections between California and Majorca related to ALSD. A relevant publication was found in which it described a visit to the Central Valley of California in August 1993 of a group of main stakeholders of the Balearic almond sector (agricultural extensionists, almond cooperative members, etc.) to learn about crop management in California[52]. Although there could have been other older non-documented visits and pathways of introduction, the most plausible explanation of how two coetaneous ALSD-related *Xf* strains only known at that time in California reached Majorca is that infected scions were brought from California into Majorca and grafted onto local rootstocks. To address this hypothesis, we tested whether infected almond buds could transmit *Xf* and develop ALSD. In this transmission experiment, only two of the 13 (~15%) plants grafted with buds from *Xf*-positive trees in June showed scorch symptoms on both the rootstock and the scion three months after grafting, and *Xf* could be detected by qPCR and isolated. In our grafting transmission experiment, buds were collected from infected trees at the beginning of June, when trees are mostly asymptomatic and the bacterial load in the vascular system is lower and thus may be irregularly distributed along the wood tissue, which could be one of the reasons for the number of successful infections being lower than expected. Nonetheless, a 15% transmission rate extrapolated to a common plantation of one hectare considerably increases the likelihood of *Xf* establishment in new areas. Moreover, in our further efforts to link epidemiological events between California and Majorca, we identified two orchards, one in the municipality of Consell (centre of the islands) and the other in Son Carrió, where Californian almond varieties had been previously grafted onto local rootstocks. According to a farmer in Son Carrió, the plantation began to decline several years after grafting, which led him to graft again with a local almond variety onto the Californian variety. We counted the growth rings in two wood sections of a single tree and determined 1995 ± 1 as the date of grafting. We expected to detect *Xf*-DNA in growth rings around 1995, but in one single tree analysed, the oldest ring with *Xf*-DNA was from 2006 ± 2 (Supplementary Data 4).

**The ALSD outbreak in Majorca began with the introduction of *Xf* strains from California.** To delimit the divergence between the *Xf* isolates of California and Majorca, we separately analysed the phylogenies of the *fastidiosa* and *multiplex* subspecies. Details of the de novo assembled genomes and those retrieved from GenBank that are included in the phylogenetic analyses are provided in Supplementary Data 6.

***Xf* subsp. *fastidiosa*.** In total, 2239 single nucleotide polymorphisms (SNPs) were identified from the set of 1872 monocopy core genes shared by 12 USA isolates and 15 from Majorca. Maximum likelihood (ML) phylogenetic analysis confirmed that all *Xf* isolates from subspecies *fastidiosa* from Majorca form a monophyletic cluster (1000 bootstrap scores) within the Californian clade (Supplementary Fig. 7). Similar topologies in phylogenetic trees were inferred using Bayesian analysis at the divergence time applying different combinations on the model parameters (Supplementary Table 4). Among these models, the uncorrelated relaxed exponential clock model combined with the Bayesian Skyline coalescent model produced the best-fit phylogeny. Root-to-tip dating provided little temporal signal but was significant enough to calibrate the molecular clock without tree priors (Supplementary Fig. 8a). Thus, we calculated the time of the most recent common ancestor for the Majorcan clade to be around 2004 (95% highest posterior density, HPD: 1982–2015). The average substitution rate calculated was $7.71 \times 10^{-7}$ substitutions nucleotide-site$^{-1}$ year$^{-1}$ (95% HPD: $1.20 \times 10^{-7}$–$1.69 \times 10^{-6}$), almost identical to that reported previously[12,53], and corresponding to ~1.3 SNPs core-genome$^{-1}$ year$^{-1}$.

Although these and other phylogenetic trees built without restrictions at clade nodes and tree roots identified clades from the eastern United States, California and Majorca, they all showed wide 95% HPD bars at their root heights and internal nodes. Such uncertainties were mainly due to the variation in the substitution rates in the branches due to the small number of genomes that covered the entire timespan of the divergence analysis[12]. To overcome this drawback, we introduced strong informative tree priors to our model based on the data presented herein and in other published works[54,55] (Table 1). Thus, we legitimately placed 1998 as the minimum age limit of the Majorcan monophyletic clade (our equivalent of a fossil record in an unequivocally well-dated stratum) and 1993 as the expected node age with a normal logarithmic distribution (the expected introduction event). In addition, we limited the minimum age for the tree root to 1892, the year Pierce's disease was first reported in Southern California[55]. Once we entered these tree priors into the model, we estimated the time of the most recent common ancestor for the Majorcan clade to be 1994 (95% HPD: 1990–1997), close to the expected introduction event in 1993. The closest relative to the Majorcan clade was the strain CFPB 8071, isolated from an almond tree in California, which diverged around 1983 (95% HPD, 1976–1986; posterior probability, pp = 1). Our time-calibrated phylogenetic tree correctly assigned the dates for the interior nodes of each clade related to the chronology of the appearance of ALSD in California[9] and the estimated date of 1882 (95% HPD: 1878–1886; pp = 1) for the introduction of *Xf* ST1 in the USA[55] (Fig. 5).

Despite the lower number of sequenced isolates ($n = 6$), the California clade showed a greater genetic polymorphism (1074 SNPs; nucleotide diversity, $\pi = 2.6 \times 10^{-4}$) and a deeper phylogenetic structure ($\pi = 1.8 \times 10^{-6}$) than the 16 SNPs found within the Majorcan clade ($n = 15$) and its shallow comb-shaped clade topology (Fig. 5). This tree pattern is expected for the introduction of a small subset of genotypes from a larger source population into a new area (founder effect) after undergoing genetic drift[56]. Such differences in the number of SNPs between both transcontinental populations also reflect a longer timespan from the introduction event[55] (~1892) and the largest geographic area in California (423,970 km$^2$) with respect to the recent introduction in Majorca (~1993) and its limited geographical range (3640 km$^2$). On the other hand, the presence of two subclades within the Majorcan clade suggests some incipient evolutionary divergence from the introductory event. The genetic variability between and within these subclades was compared by calculating the fixation index (Fst), which measures population differences due to genetic structure. A higher Fst (0.21) than

**Table 1 Prior distribution used to calibrate node ages in BEAUTI for the Bayesian phylogenetic analyses of *Xylella fastidiosa* subsp. *fastidiosa* and *multiplex*.**

| Tree/clade/root | Distribution | Parameters[a] | Reference |
|---|---|---|---|
| *Xf* subsp. *fastidiosa* | | | |
| Clade Majorca | Lognormal | offset = 20 (1998); μ = 5 (1998–1993); SE = 2.0 | This study |
| Tree root | Lognormal | offset = 126 (1892); μ = 10; SE = 3.0 | Pierce[55] |
| *Xf* subsp. *multiplex* | | | |
| Clade (Majorca(Men)) | Lognormal | offset = 18 (2000); μ = 7 (2000–1993); SE = 3.0 | This study |
| Clade Alicante[b] | Normal | μ = 20; SE = 4.0 | This study |
| Clade Corsica[c] | Normal | μ = 25; (2001 + 1985)/2 = 1993; SE = 4.0 | Soubeyrand, et al.[54] |
| Clade California[d] | Lognormal | offset = 7(2011), μ = 16; SE = 4.0 | Supplementary Data 6[e] |
| Tree root | Normal | offset = 23 (1995), μ = 10; SE = 4.0. | Supplementary Data 6[e] |

[a]Time 0 is defined as 2018 (calendar year); μ = mean of the distribution; SE are increased according to the uncertainty of the clade prior.
[b]To the best of our knowledge, no problems of almond mortality had been reported before 2017 in Alicante; we therefore assumed a similar survival time for almonds as in Majorca and used a relaxed distribution to accommodate uncertainty in the prior.
[c]Strain CFBP8416 from Corsica was included in the California clade because no other isolate was available of ST7 in Corsica. We used the two introduction scenarios proposed by Soubeyrand et al.[54] to estimate μ.
[d]Strain Dixon was included in the Corsica clade as it was genetically very close to isolates CFBP8417 and CFBP8418.
[e]The time of the youngest tip date of the isolate within the clade was used to define the offset; μ is calculated as the time between the offset and the time of the oldest isolate within the clade.

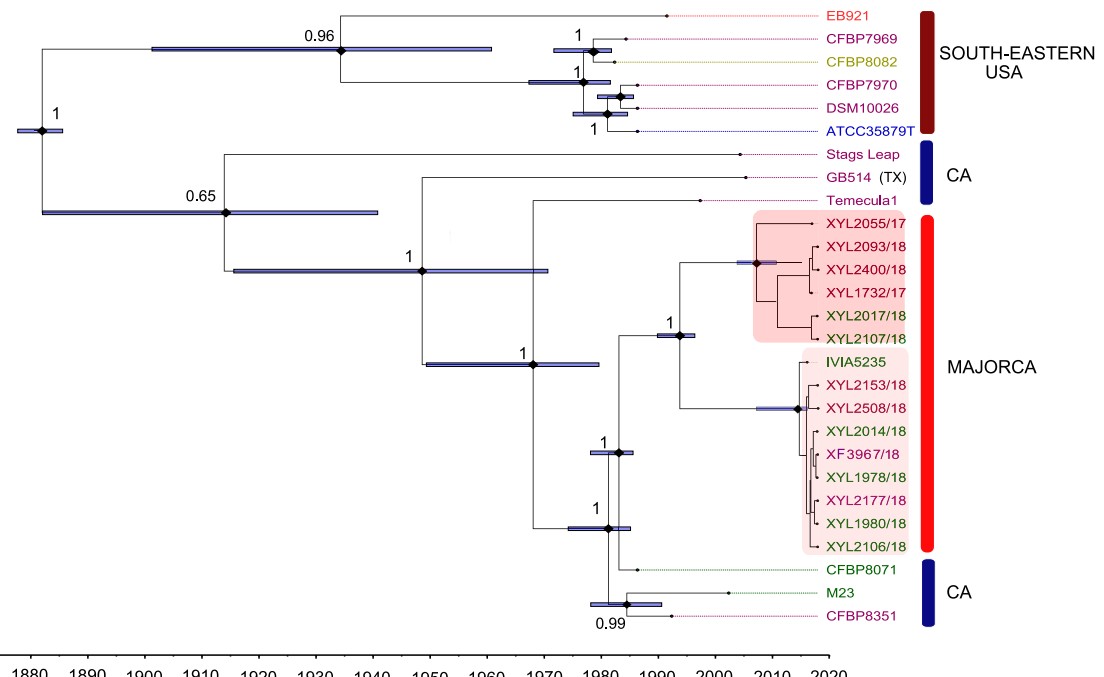

**Fig. 5 Time-calibrated maximum clade credibility tree based on core-genome sequences for ST1/ST2 isolates of *Xylella fastidiosa* subsp. *fastidiosa* from Majorca, California and the South-eastern USA.** Informative tree priors used to calibrate the nodes are given in Table 1. The Majorcan clade forms a monophyletic cluster nested within the Californian clade *sensu lato*. The tips (isolate names) are coloured according to the host: purple = grapevine; green = almond; red = elderberry; golden = *Ambrosia artemisiifolia* and blue = coffee plant. Isolate GB514 from Texas falls within the Californian Clade. Node bars = 95% highest posterior density (HPD) node age; node number represents the posterior probability.

expected for a founder introduction was obtained. However, contrary to expectations, both subpopulations were apparently not genetically structured either by their hosts or by their geographical distribution, since the isolates of vines and almonds throughout the island were interspersed within both subclades (Fig. 5). Furthermore, when the Majorcan isolates were grouped according to the host and analysed for their genetic diversity within and between each group (vines and almond trees), we found an Fst = ~0. These data strongly suggest the existence of cross infections between almond trees and vines by ST1 strains of subspecies *fastidiosa*, a limited gene flow between both subclades and a high clonality within them, since no recombination was

detected using ClonalFrameML[57] (Supplementary Fig. 9). Before drawing conclusions about the genetic structure of both subclades, more samples would need to be included in the genetic analysis. Nonetheless, these preliminary data agree with our hypothesis of the introduction of a single or few *Xf* ST1 genotypes through infected buds or scions and their subsequent long-distance spread through grafting across the island.

***Xf* subsp. *multiplex*.** Differences among ST81 isolates were limited to 11 SNPs of the core genome (1,664,716 base pairs), a range similar to that found in the ST1 population in Majorca (independence test; χ = 0.42, P = 0.52). Furthermore, both subspecies

showed a similar nucleotide diversity (subsp. *multiplex*, π = 2.5 × $10^{-6}$), which supports the idea that they have undergone parallel demographic trajectories since the time of their introductions. As also occurred with ST1, the same haplotypes of ST81 were detected in two hosts, in this case on almonds and wild olive trees. Cross infections between both hosts need to be tested in inoculation experiments to confirm whether the widespread distribution of wild olive trees has contributed to the spread of ALSD across the island (see Supplementary Fig. 1), as well as the possible spread to 12 other hosts infected with ST81, including the fig and the ash of narrow leaves. Identical haplotypes have also been found on the islands of Majorca and Menorca, which are interspersed in the same clade in the phylogenetic analysis, indicating that the subpopulation of the island of Menorca very possibly derives from an introductory event from the main island. Hereafter, we denote this clade formed by ST81 isolates as (Majorca (Menorca)). Possible routes of entry include the transfer of infected plant material for grafting or the introduction of infected insects into ships, vehicles or containers that move daily between Majorca and Menorca.

The concatenated sequence alignment of 25 core genomes comprised 1,615,082 nucleotides. A total of 2511 SNPs (544 singletons, i.e. mutations appearing only once among the sequences) were identified, of which 1341 SNPs (π = 4.1 × $10^{-4}$) belonged to the Californian clade *sensu stricto*. ML analysis confirmed that the isolates of the *multiplex* subspecies from (Majorca (Menorca)), Alicante and Corsica, together with isolates belonging to ST6 and ST7 from California, form a robust monophyletic group previously described by Landa et al.[31] (Supplementary Fig. 7b). Because all European isolates were collected between 2015 and 2018 (n = 18), only a weak temporal signal was obtained that incurs pseudo-replication to calibrate the molecular clock for the panel of 25 genomes (Supplementary Fig. 8b). To solve this problem, we divided the panel into four monophyletic clades, i.e. California, (Majorca (Menorca)), Corsica and Alicante, based on the ML analysis (Supplementary Fig. 7b) and ran the best-fit model (Supplementary Table 5). For each monophyletic clade, we calibrated the age of the nodes based on the tree priors and transferred their uncertainties to the probability distribution (Table 1). For example, we used an intermediate introduction scenario (1993) for the Corsican clade due to the lack of references on the *Xf* strains for each of the scenarios in the mathematical model, as well as the few genomes available to infer the time of introduction[54]. For the clade (Majorca (Menorca)), DNA detection of the *multiplex* subspecies in a ring dated 2000 ± 2 in one almond tree in Majorca (Supplementary Data 4) provided a minimum limit for the age of this node. On this basis, it was estimated that the time of the most recent common ancestor of ST81 in the Balearic Islands would be 1995 (95% HPD: 1991–1998) and the introduction of Alicante ~2005 (95% HPD: 1996–2017) and the isolates CFPB8417 and CFPB8418 of Corsica ~2000 (95% HPD: 1995–2002) (Fig. 6). These results are consistent with our proposed introduction event in 1993 and are also in line with the current epidemiological stage of ALSD in Alicante, while supporting one of the introduction scenarios proposed for Corsica in 2001 by Soubeyrand et al.[54], and the age of the nodes agrees with the chronology of the appearance of the ALSD in California[9].

## Conclusions

Our research reduces the date of the first *Xf* outbreak in a crop in Europe by 20 years, in line with the predictions proposed for an ancient introduction of *Xf* in Corsica based on mathematical models[54]. It also explains how ST1, the strain that causes Pierce's disease of grapevines, in addition to ALSD, reached Majorca

several years before being regulated as a harmful organism in the Annex list of the Council Directive 2000/29/EC. Taken together, our epidemiological, dendrochronological, climatic and molecular data shed light on the role of *Xf* in almond decline and its implication in the hitherto overlooked ALSD outbreak. The pathogen remained undetected for decades in the Mediterranean basin in almond or other hosts, confused for other pathogens or for symptoms of drought stress. This study collects many pieces of evidence that point to 1993 as the most likely date for the arrival of *Xf* in Majorca, though we cannot rule out unlikely previous introductions. We believe that the introductions from California to Majorca of the strains belonging to ST1 and ST81 were coetaneous or close independent events in time, probably through material infected with both lineages. This hypothesis is supported by (i) a similar number of SNPs accumulated within their populations—suggesting that they would have experienced parallel demographic trajectories—, (ii) the close dates assigned in the time-calibrated phylogeny for both STs and (iii) the fact that coinfections do not seem rare in areas from California[40,41] and Majorca (Supplementary Data 4). Our findings also confirm that other invasion routes in addition to the ornamental plant trade have occurred in Europe. Emulating the successful history of almond cultivation in California, crop improvement programs were implemented in the main almond-producing countries in the Mediterranean and the Near East in the 1980s and 1990s. Finally, our reconstruction of ALSD in Majorca has important implications for European plant health policy by questioning the adequacy of eradication and contingency strategies without any prior investigation into the epidemiological history of the outbreaks that are intended to be contained.

## Methods

**DNA extraction and molecular assays**. Plant samples from different sources, i.e. *Xf* official surveillance, cooperatives, farmers, agricultural extension services, etc., were received and processed in the laboratory for *Xf* qPCR analysis and registered in the database of the Official Plant Health Laboratory of the Balearic Islands (LOSVIB). DNA extraction from plant extracts was performed using an EZNA HP Plant Mini kit (Omega-Biotek, Norcross, Georgia, USA) based on CTAB, as described in the EPPO protocol[15], whereas *Xf* genomic DNA was extracted using the Wizard DNA Promega kit. DNA extracts were tested for the presence of *Xf* by qPCR using two species-specific protocols[15] with primers XF-F/XF-R and the dual-labelled probe XF-P (Harper et al.[45]) and primers HL5/HL6 and a TaqMan probe (Francis et al.[46]) using an ECO thermal cycler (PCRmax, Staffordshire, UK). DNA from *Xf* subsp. *pauca* strain CoDiRo or *Xf* subsp. *fastidiosa* strain Temecula were used as positive controls. Additionally, for some experiments and for identification of the *Xf* subspecies infecting the trees, a duplex TaqMan qPCR assay that differentiates *Xf* isolates belonging to subspecies *fastidiosa* and *multiplex*[58] was performed using a LightCycler® 480 Instrument (Roche, Madrid, Spain). Colour compensation was applied to minimize potential crosstalk between fluorophores according to the operator's manual. Finally, MLST analysis amplified seven housekeeping genes[59] using DNA extracted from plant tissue, bacterial suspensions or insect samples to identify the subspecies and, when possible, the ST of *Xf* in selected samples. We also determined the ST of the isolates from pre-assembled genomes by retrieving the sequence of the seven loci used for typing *Xf*, as described by Larsen et al.[60], using the *Xf* MLST Database (https://pubmlst.org/xfastidiosa).

**Xf isolation**. *Xf* isolates were obtained from samples of almond trees, vines and other hosts in the LOSVIB in accordance with the EPPO protocol[15]. We selected for the isolation leaf samples that had low Cq values in qPCR and were sampled between June and September 2017 and 2018. We used periwinkle wilt-GelRite (PWG) medium prepared as described in the EPPO PM7-24[15] for isolation and subculturing. Three 10-ml drops of the homogenized leaf extract were placed in a Petri dish, streaked on PWG, sealed with Parafilm and incubated upside down at 28 °C in darkness. Candidate colonies were transferred onto fresh PWG, and their identity was tested by qPCR as described by Olmo et al.[27]. Single colonies were triple-cloned on PWG, stored in PW broth with 20% glycerol and kept at −80 °C.

**ALSD progression and incidence**. To determine whether the development of ALSD symptoms, shoot and branch diebacks, and subsequent tree death followed an orderly sequence in time, we used panoramic images from the Google street view repository for the summers of 2009, 2011, 2012, 2013, 2014, 2016, 2017 and 2018.

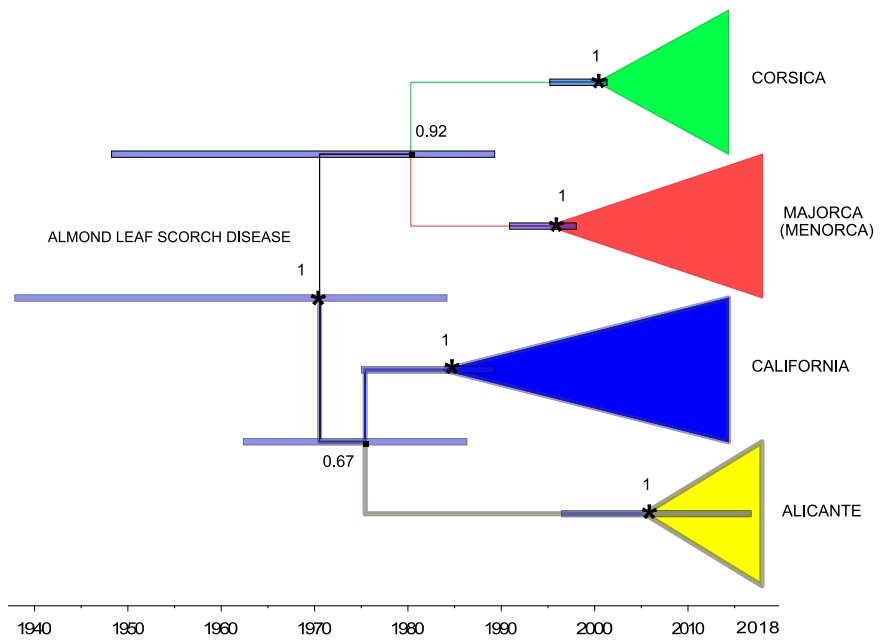

**Fig. 6 Time-calibrated maximum clade credibility tree from a combination of two runs of 500 million states and 50 million burn-ins based on core-genome sequences for isolates of *Xylella fastidiosa* (*Xf*) subsp. *multiplex* from Majorca, Alicante, Corsica and California (including isolate Griffins from Georgia, USA).** The node of the mid-root tree represents the source population from California from which the European clades are derived. Strong informative priors are described in Table 1. were used to calibrate the clade nodes. The model is congruent with the temporal expansion of the almond leaf scorch disease (ALSD) in California (1930s–1980s[9]), the likely introduction of *Xf* ST81 in Majorca around 1993, and the time of introduction in Corsica proposed by Soubeyrand et al.[54] and the more recent epidemic of ALSD in Alicante (Spain). Node bars = 95% highest posterior density (HPD) node age; node number represents the posterior probability. Subtrees of each clade were collapsed to mask artificial long branches due to trade-offs between tree prior node ages and substitution rates.

Trees falling along 1-km roadside transects were examined in various areas where the yearly sequence of images was available. Observations were ranked into 5 categories: 0 = healthy trees with no symptoms of ALSD; 1 = a branch or the whole canopy with leaf scorch symptoms; 2 = shoot and branch dieback affecting between 1 and <25% of the canopy; 3 = 25–75% dieback; and 4 = >75% dieback or dead trees (Supplementary Data 3). We used a Friedman One-Way Repeated Measure Analysis of Variance by Ranks and Kendall concordance to test whether ALSD symptoms preceded diebacks symptoms attributed to fungi following a severity scale sequence (Supplementary Fig. 3).

To estimate the ALSD incidence, i.e. % infected plants in a population, minimizing disease heterogeneity among orchards (<2 ha), we performed cluster sampling. Overall, 126 almond orchards (sampling units) distributed along 36 municipalities of Majorca were surveyed between August and September of 2017 (Supplementary Data 1). We excluded irrigated orchards, as they constitute a negligible fraction (<3%) of the predominant rainfed plantations. Orchards in each municipality were selected depending on the availability of a safe place along the roadside to park the car and georeferenced with a mobile phone. From the edge of the orchard, 30 trees were visually examined within three to five randomly selected rows, and the number of symptomatic (including diebacks) and dead trees was recorded. Because common perception among farmers was that older trees were more affected than younger plantations, we tested whether age had an effect on disease incidence and mortality. The orchard age was visually estimated based on the trunk diameter and classified as young (≤30 years) and old (>30 years) plantations. To validate our visual incidence assessment, we surveyed all the trees of a 25-year orchard by walking (n = 192 trees).

We used repository panoramic images from Google street view as a proxy to assess the disease incidence and mortality in 2012 around Majorca. This approach has been used to assess the distribution of the pine processionary moth in France[47]. In 2012, Google street view panoramic photographs taken between May and October covered extensive areas of the island. Points of observation were successively selected from satellite images to cover all the areas of distribution of almond plantations in Majorca. Unfilled areas were zoomed in on until almond plantations were visualized next to roads. We selected a road point and progressively zoomed the satellite image until it switched to a panoramic ground image. We then virtually drove the car a few metres along the road to search for optimum panoramic images to visualize the almond orchard from where we could examine 30 trees close to the camera (less than 100 m away) and recorded incidence, mortality and tree age as described before (Supplementary Data 2). To minimize observational bias, all observations were performed by the same operator who was trained in the 2017 field assessment.

The georeferenced data on ALSD incidence and mortality among orchards obtained in the field assessment in 2017 (Supplementary Data 1) and from Google street view in 2012 (Supplementary Data 2) were mapped using QGIS software[48]. To determine if the distance from the supposed main focus of "Son Carrió" or the tree age could have any effect on the incidence and mortality of the disease, we used a generalized linear model, as implemented in the lme4 package[61] in R[62], in which the distance from the hypothetical focus (<20 km; >20 km) and tree age (<30 years; >30 years) were treated as fixed effects. We used a 20 km threshold as a reference for the approximate radius of dispersion due to transmission by vector insects in a given area as opposed to long-distance dispersal due to grafts or stowaway insects in vehicles[63]. The disease incidence assessment data for 2017 and 2012 were also used to estimate the statistical dependence between the rankings of ALSD incidence and mortality among orchards using Spearman correlation.

**Sampling for *Xf* DNA in growth rings**. In the winter of 2017, we realized that wood samples from *Xf*-infected deciduous trees could be used to detect past infections, regardless of the season of the year. We reasoned that *Xf* would over-winter in the xylem tissue from the previous year. Because axial water and nutrient flows in the wood of temperate ring-porous species are restricted to the last, or last few, annual rings[64,65], we assumed for *Xf* an annual directional movement following this gradient (Supplementary Fig. 5). Under this assumption, living bacteria would colonize the new vascular tissue formed in the spring that interconnects the newly formed leaves, while the dead bacterial cells would be left behind in last year's wood. The seasonal growth cycle in vascular tissue was confirmed for three years by looking at the monthly frequency distribution of positive qPCRs in almond samples (Supplementary Fig. 5a). We also predicted, according to the assumptions of our model, that the concentration of *Xf*-DNA measured as Cq values in the qPCR would increase in the direction of the pith towards the bark, i.e. a lower Cq in the wood of the periphery. We used this spatial trait to date the presence of *Xf* DNA by qPCR in the annual growth rings of felled almond wood.

According to the Commission implementing Decision (EU) 2015/789, all *Xf*-positive trees must be destroyed. We took samples of 34 positive *Xf* almond trees before they were burned to analyse the chronology of infections and infer any spatial structure from the spread of the disease in Majorca. For all trees, georeferenced coordinates were taken. Three 10-cm-height and chest-height-diameter (DBH) cross sections and two branches were cut with a chainsaw and stored in labelled bags. In the laboratory, the wood sections were first levelled with a manual electric planer and then sanded with progressively thinner grids, rinsed in running water with a drop of soap, and air dried. The 20 outer growth rings were

marked under a stereoscopic microscope with different pencil colours. Using a Dremel vertical drill stand with a 0.4 mm bit, we obtained ~0.5 mg of sawdust from each ring, discarding the first particles and starting sampling from the inner rings to the outer rings to minimize possible DNA contamination. Sawdust was carefully collected and placed in a labelled bag (Bioreba) and ground using a Homex 6 homogenizer (Bioreba instruments, NY). DNA extraction and qPCR analysis were performed as described by Olmo et al.[27]. Cq values were determined to quantitatively estimate *Xf* DNA in the growth ring. Sample analysis by qPCR was performed in duplicate at the LOSVIB and IAS-CSIC facilities. In addition, samples of trees selected from the oldest wood rings were used to identify the *Xf* subspecies that infect the tree using a TaqMan qPCR assay that differentiates *Xf* isolates belonging to *fastidiosa* and *multiplex* subspecies[58].

**Survival analysis**. A Kaplan–Meier survival estimate was determined from the data obtained in the years of infection of each of the trees in the previous section (Supplementary Data 4D). At the time the trees were felled, the event of interest, tree death (decrepitude), had not occurred in 29 of the 34 trees; therefore, they were right censored in the survival analysis (i.e. survival was above that in 2017–2018 but by an unknown amount). We used the package 'survival' in R to conduct the Kaplan–Meier survival estimate.

**Inoculation and grafting experiments**. In 2018, twenty 3-year-old rootstock-stem cultivars grown in 20-L plastic pots with a standard potting soil mix were used in the inoculation trial (Supplemental Table 2). The pots were randomly distributed in rows of 12 plants along an insect-proof tunnel exposed to air temperature. The almond plants were drip irrigated daily, fertilized biweekly with a slow-release fertilizer granulate, and treated with insecticides and fungicides as needed until the end of the experiment. Two weeks before the start of the inoculation test, leaf samples were collected from all plants, and the presence of *Xf* was analysed through qPCR. For this inoculation experiment, we used the strain XLY 2055/17 (*Xf*. subsp. *fastidiosa*; ST1), whose genome had been recently sequenced[42]. The isolate was cultured in BYCE medium at 28 °C for 7–10 days, following the EPPO PM7/24 protocol[15]. The cells were collected by scraping the colonies and suspending them in phosphate-buffered saline (PBS) until a turbid suspension was obtained ($10^8$–$10^9$ cells ml$^{-1}$). Plants were mechanically inoculated by pin-prick inoculation[30] with slight modifications. A 10-μl drop of the bacterial suspension was pipetted on the leaf axil, and the leaf axil was punctured five times with an entomological needle. Eight replicates per scion-rootstock combination were inoculated with the bacterial suspension and four plants per cultivar, with a drop of PBS as a control. Inoculation was repeated two weeks thereafter by piercing the next leaf axil above that previously inoculated. Symptomatic and asymptomatic plants were tested by qPCR for *Xf* infection at 16 weeks post inoculation, taking the petiole of the second and fifth leaves above the second point of inoculation. Five leaves of all the inoculated plants were used for *Xf* isolation, following the EPPO protocol[15]. In October 2017, five almond varieties (Jordi, Guara, Marta, Masbovera, Filau and Vivot) were inoculated with isolate XYL1752/17 of *Xf* subsp. *multiplex* ST81 using pin-prick inoculation method[30], and infections tested in August 2018.

We also tested whether *Xf* could be transmitted by bud grafting, as previously demonstrated[9]. Buds were taken from branches of an almond tree infected with *Xf* (sample XYL 1699/18; qPCR positive using Harper et al.[45] and Francis et al.[46] protocols) in June 2018. Two buds per plant were grafted onto 2-year-old rootstocks, which had previously been tested for *Xf* infection-free and stored in an insect-proof greenhouse. These were examined in September 2018 for the presence of *Xf* symptoms, and *Xf* infection was confirmed by qPCR, as described above.

**Natural infection of living germplasm collections**. In the summers of 2017 and 2018, we monitored the incidence of ALSD in four collections of live almond germplasm banks distributed throughout the island. The almond varieties in these germplasm collections are well catalogued and can therefore be used to establish susceptibility to *Xf*. Because the presence of *Xf* went unnoticed until 2016, these varieties have been naturally exposed to the pathogen for several years. The symptomatology of ALSD was recorded, and samples were taken during the summer for qPCR analysis.

**Review of fungal trunk pathogens implication in almond decline**. We re-examined publications on the involvement of fungal trunk pathogens in almond decline in Majorca[1–3,49] to compare them with fungal species implicated in 'olive quick decline syndrome' in Italy[50,66,67]. We also collected data from Olmo et al.[49] work to determine whether the regional diversity of fungal species associated with almond decline could be related to *Xf* infections.

**Climatic data**. To decipher whether water stress, as one of the main symptoms of the decline of almonds, was induced by climatic or biological factors, we analysed the monthly water balance of the almond crop by collating the climatic data between 1988 and 2017 from nine meteorological stations in Majorca at AEMET (aemet.es). The monthly potential evapotranspiration (ET$_0$) was calculated using the Hargreaves[68] equation ET$_0$ = [0.0023 (tmed + 17.78) R$_0$ * (tmax-tmin) 0.5] * 30. The annual CWDi was calculated during the growing season from February to August as CWDi = Σ (ET0-Ppn) w, where Pp is the contribution by rain and 'w' is

a monthly weight factor (0.05; 0.1; 0.2; 0.3; 0.2; 0.1 and 0.05), which takes into account the development of the canopy leaf area throughout the season (Supplementary Data 5). We performed an independent two-sample Student's *t*-test to determine whether there had been an intensification in drought during the 15-year timespan after 2003 compared to droughts before 2003.

**Transmission tests**. Nymphs of *P. spumarius* were field-collected from weeds and raised to adults on potted plants of *Argyranthemum frutescens* (Asteraceae), *Ocymum basilicum* (Labiatae) and *Foeniculum vulgare* (Umbelliferae) in a greenhouse. Due to the lack of of transstadial or transovarial transmission of *Xf* in spittlebugs[8], we assumed that all nymphs were free from the bacterium. Transmission of the bacterium by *P. spumarius* was carried out by exposing the adult insects for 4 days (acquisition access period (AAP)) to infected almonds that had previously been positive for qPCR. After AAP, insects were divided into groups and caged with an almond branch from a plant that had previously been found to be infection-free (asymptomatic and PCR negative) during an inoculation access period (IAP) from 96 h. Controls consisting of plants exposed to uninfected insects as well as plants without insects were included in the experiments. All insects were removed from the plants at the end of IAP, killed by cooling and freezing, and stored in ethanol at 96 °C for subsequent qPCR analysis. After 2–3 months, symptoms were recorded, and the presence of bacteria in plants and insects was confirmed by qPCR analysis. In positive cases, an attempt was made to isolate the bacteria from the infected plant[15].

**Whole-genome sequencing and phylogenetic analyses**. Whole-genome sequencing of 16 isolates obtained in this study was performed using the Illumina HiSeq 2000 with paired-end reads of 150 bases. Illumina reads were assembled de novo using SPAdes 3.6.0 under default parameters. We analysed the genomes belonging to *fastidiosa* and *multiplex* subspecies separately. For the first, we included in the phylogenetic analysis 12 genomes of isolates from the USA (ST1/ ST2) available at GenBank in December 2019, along with 15 newly sequenced ST1 genomes obtained from different host plants in Majorca. To catch the more recent evolutionary history of the *Xf* subsp. *multiplex* isolates related to ALSD, i.e. the last 100 years, without increasing the timescale, we included all genomes available at GenBank in December 2019 from Europe and isolates from California used by Landa et al.[31], except those from Tuscany due to their relatedness to older east-USA *multiplex* lineages, along with four isolates from Majorca and Menorca sequenced in this work (Table 1). All genomes were annotated with PROKKA for comparison[69], and the protein amino acid sequences were compared with the GET_HOMOLOGUES software[70], using the criterion of 50% similarity over 50% coverage alignment. Core-genome analysis was performed with three different clustering algorithms, bidirectional best-hits (BDBH), COGtriangle (COG), and OrthoMCL (OMCL). The nucleotide sequences of the monocopy genes from the core genome were aligned using MAFFT v7[71], and the results were concatenated into matrices of 1,713,585 bp (*Xf* subsp. *fastidiosa*) and 1,615,082 bp (*Xf*. subsp. *multiplex*) lengths that were manually curated to ensure high-quality data for subsequent phylogenetic analyses. The substitution rate, emergence times and divergence dates were estimated using Bayesian approaches as implemented in BEAST v1.10.4[72]. The best evolutionary model was estimated with PartitionFinder 2 using the Bayesian information criterion[73]. All possible combinations of three different molecular clocks (strict, exponential-relaxed and lognormal-relaxed) and two tree models (constant size coalescent and Bayesian skyline coalescent) were compared using Bayes factors based on marginal likelihood estimations through path sampling and stepping-stone calculations in BEAST[74]. Each analysis consisted of two independent runs of 500 million generations sampled every 10,000 states under the GTR + I + G (*fastidiosa*) or HKY (*multiplex*) substitution model. Tree calibration was performed by assigning the isolation date to each tip of the tree (month and year). MCMC convergence was assessed in Tracer v1.7.1, ensuring effective sample size values higher than 200. The best-fit model was selected to include priors on node time (tree prior) adjusting the probability distribution of the node age based on data obtained in this research and the introduction scenarios for Corsica proposed by Soubeyrand et al.[54]. Because no distinction was mentioned between *Xf* strains in the model proposed for Corsica, we opted to include a middle introduction scenario (1993) and a normal distribution to incorporate uncertainty. For the used dataset, the exponential-relaxed clock combined with the Bayesian skyline model provided the best fit. Log-Combiner v1.10.4 was used to discard 10% of the states as burn-in and to combine the trees from the two independent runs. Finally, a maximum clade credibility tree from the combined trees was obtained using TreeAnnotator v1.10.4.

To understand the recent evolutionary history undergone by the *Xf* population in Majorca and Menorca in the context of their source populations in California, we used the whole-genome sequence data to calculate descriptive statistics with DnaSP6[75].

**Statistics and reproducibility**. All statistical analyses were performed using computing language and environment R[62], except the descriptive analysis of genetic populations, which was conducted with DnaSP6[75]. In each of the Methods sections, we mentioned the statistical test and the R functions and packages used.

Full details of the R scripts for each of the statistical tests are provided in the Supplementary Software 1.

**Reporting summary**. Further information on research design is available in the Nature Research Reporting Summary linked to this article.

## Data availability

*Xf* MLST profiles and sequences obtained from insect and plant samples from this study are available at figshare[76] (https://figshare.com/s/0b571280bcd7738a524d; 10.6084/m9.figshare.12378302). All of the raw data used in this publication for figures are available both in the Supplemental Data and in the figshare online repository[77] (https://figshare.com/s/a4a78390e02e8cc123d4;10.6084/m9.figshare.12488342). Raw genome sequencing data used in this study have been submitted to Sequence Read Archive under the following accession numbers: SRR11931324- SRR11931339.

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

## Acknowledgements

This work was partially funded by projects XF-ACTORS (*Xylella fastidiosa* Active Containment Through a Multidisciplinary-Oriented Research Strategy; grant 727987 from European Union's Horizon 2020 Framework Research Programme) and E-RTA2017-00004-02 and E-RTA2017-00004-04 (Desarrollo de estrategias de erradicación, contención y control de *X. fastidiosa* en España) from 'Programa Estatal de I + D + I Orientada a los Retos de la Sociedad of the Spanish Government' and FEDER) and from the Organización Profesional del Aceite de Oliva Español'. M.P.V-A. was recipient of a PhD fellowship from Intramural Project 201840E111 from CSIC. A.B. was recipient of a postdoctoral UIB contract funded by the Vicerectorado de Investigación e Internacionalización of the University of the Balearic Islands.

## Author contributions

E.M., B.B.L., D.B., D.O. and M.G. designed the research; A.B., A.J., A.N., A.P., B.B.L., B.Q., D.B., D.O., E.M., F.A., G.S., J.A.J., J.D.G., M.G., M.M., M.P.V., O.B. and P.A.G. performed the research; B.B.L., E.M., J.A. J., M.G. and M.M. analysed the data; E.M. wrote the paper; and B.B.L. revised the paper.

## Competing interests

The authors declare no competing interests.
