## [Peer Review File · Communications Biology]

Reviewers' comments:

Reviewer #1 (Remarks to the Author):

The authors describe the analysis of *Xylella fastidiosa* (Xf) as the causal agent of almond scorch disease (ALSD) in Mallorca, describing the potential distribution and likely entry timescale. Furthermore, the authors carried out population analysis to compare the Xf isolates identified in Mallorca to those identified in California and state that due to the close genetic proximity etc. that the Mallorca Xf was directly introduced from California in the early 1990s. They describe further details of the Xf substructure in Mallorca through MLST analysis, indicating that it represents two distinct subspecies. They also utilised Xf analysis in tree rings to provide the likely date of Xf introduction.

General comments:

Overall the study addresses an important question regarding the likely source of the Xf ALSD introduction into Mallorca, whilst considering a pathogen that is of great broad interest. However, this current study also builds on previous studies that should be more clearly highlighted to provide context. This includes a study by the authors (Landa et al., 2020) that used sequence analysis of two Xf strains from Mallorca (one from almonds) that already states that Xf was likely introduced into Europe (and Mallorca) from California due to genetic proximity. Furthermore, the proposal of Soubeyrand et al. (2018) describing two likely introduction periods into Europe, one of which fits well with the current study should be better discussed. Overall, the manuscript is very difficult to follow in its current format and thereby the major finding is not clear. Although interesting data is presented it is difficult in the current format to assess how this contributes to that that was described previously. For instance, the main aim is not really clear to me. It seems that determining the origin may have been the main aim? But this is described in the previous publication.

The format, complex nature and specificity of the terminology would be extremely difficult for a general audience to follow. There are many terms that are not described, and the logic is not easy to follow. The lack of referencing in many places also makes it difficult to determine whether certain findings are from this or previous studies. In its current format it would be much more suited to a subject specific pathology-focused journal.

The manuscript would greatly benefit from specialist editing to improve the general English quality and flow. I have not provided any English language corrections below as the manuscript requires extensive editing. Ensuring the aims of the study and hypothesis are stated in both the abstract and throughout would help to better guide readers.

It would be beneficial to include all methods in the main text.

All data should be deposited in a public repository and not available "upon reasonable request", see detailed comments. This is essential to support reproducibility.

Specific comments:

Abstract

Line 30: "knowledge on their pathways and chronology of entry remains circumstantial". How is the pathway circumstantial when this previous study stated also that it likely was introduced from California (Landa et al., 2020)?

Line 41: "Pierce's disease of grapevines" this is only introduced in last sentence with no context.

Introduction

Line 43-44: Please add a reference.

Line 53: Reference required.

Line 70: Reference required to the annual survey data.

Line 93: Here this should again refer to (Landa et al., 2020). Stating that "To disentangle the likely origin of Xf epidemics in Mallorca" gives the impression this is unknown and has not been studied previously.

Line 105: Is there a reference to these comments from plant health advisors or at least names of who provided this information?

Line 111: It is not clear what 'almost identical' means and whether this is significant in this context.

Results & Discussion

Lines 132-133 & 264-265: These statements seem unnecessary and could be incorporated into the text to improve the flow of the text.

Lines 150-155: How was the mean disease incidence assessed?

Lines 155-156: How did you estimate the number of remaining dead trees?

Line 170: Can the spreading centres be indicated on the Figure.

Line 171: This section then jumps to tree ring analysis. Can the authors describe this analysis better i.e. the justification, hypothesis & how it was carried out? This would make it easier to follow.

Lines 179-180: An alternative explanation could be a higher degree of susceptibility.

Line 181: How was the GMSV used, it would be useful to provide the dates in the results section. Is there a study that has used this in a similar manner that could be referenced as I could imagine many issues with analysing 2D images of trees for symptoms? Is there a summary of this data that could be provided?

Lines 193-194: How many samples were analysed?

Lines 190+ & 260-262: MLST data should be deposited in a public repository.

Lines 249-262: This section is particularly difficult to follow.

Lines 267+: The link between ALSD and climatic conditions needs to be more clearly described. This section is also difficult to follow and I'm not sure of the conclusion here. Furthermore, the authors state that they have developed AWDi but it is not clear what this is or how the analysis was carried out or which statistical test was performed.

Lines 288-292: Where is this data?

Lines 292-296: Again, it is not clear where the data is that these statistics are based on? The figure panel referred to is an image rather than data.

Lines 306+: This then jumps back to a much more detailed discussion of the tree ring dating. It would be easier to follow if this analysis was discussed in a single section.

Line 318: 'Cq = 2 9' is this 2 or 29? If 2 then that would indicate a likely problem with the qPCR?

Lines 319-320: Specific primers were utilised. Were these designed in this study or a previous study? If here then please provide a table with the sequences, if not then please provide the reference.

Line 325: 'right-censored' is used multiple times but without a clear explanation. I am not sure what is meant by this term.

Lines 363-365: A total of 2/13 trees showed symptoms after grafting - this doesn't seem particularly high but is used as evidence that grafting from Californian trees was the cause of pathogen introduction?

Lines 352 & 366-368: If median survival estimates are 14 years then how were trees still alive to analyse 25 years after grafting? This would be useful to discuss.

Lines 376-386: With all Mallorca isolates being recent how did the authors use tip dates? These would surely reflect the older US population that has a higher level of diversity? This limitation is mentioned to a degree much later in lines 451-454 but it is important to highlight within this section.

Line 405: Why is the date stated (1882) outside of the confidence intervals?

Lines 406-410: As stated n=6 is a very small number of genetic analyses for the Californian group. The authors should also provide the number of samples for the Mallorcan clade.

Lines 420-422: How was genetic diversity within isolates determined from the FST value?

Lines 474-475: How do your findings reveal the invasion route? If referring to the grafting onto local rootstocks this was already hypothesised in (Landa et al., 2020) and it would be useful to understand how the current findings support this?

Lines 482-486: Here the authors state that it is remarkable that their results support an earlier introduction of Xf, but this seems to have been hypothesised by (Soubeyrand et al. 2018) in France and it would be advantageous to mention this here.

Lines 466+: The concluding remarks are very long, and it would be helpful to have a more concise summary of the major findings and their implications.

Line 532: The authors state that they didn't use all published samples, why was this? An explanation in the text would be helpful.

Line 558: How many is 'most' trees?

Methods

Lines 534+: How were the genomes mapped to the reference, SNPs called etc.

Lines 596-597: All raw sequence data must be deposited in a public repository not available "upon reasonable request". Furthermore, one supplementary file contains accessions for the assemblies used. However, whilst the assemblies are available, the WGS reads are not. These should be added to the SRA or a similar database to enable the results of the study to be verified by others as reproducible and for use in future studies.

Tables & Figures

Figure 1: Add y-axis label to panel A.

Figure 2: Add distance scale bar and label to the heat map colour bar. Overall, this figure is quite unclear and additional labels could be helpful.

Figure 3: Adding keys would be helpful for the reader.

Figure 5: Having the outline of California in the background is distracting and the authors may consider removing.

Figure 6: As above regarding the California outline.

Supplementary Data 1: Why is the age of trees only given as younger or older than 30 yrs? With the distance from Son Carrio, why is this more or less than 20 km instead of the exact value?

Supplementary Data 1: It would be useful to separate these two spreadsheets into two separate data files.

Supplementary Table 2: It would be beneficial to provide the data for each variety so it can be cross-referenced with Table 1.

Supplementary Figure 2: For the non-specialist it would be useful to briefly describe the differences in the legend.

Supplementary Figure 3: Appears to have been distorted.

Supplementary Figure 4: Do add a label to the heat map key.

Supplementary Figure 5: Please add a legend for panel B. How are the bootstraps illustrated here?

Supplementary Figure 7: The scale on the trees are missing.

Reviewer #2 (Remarks to the Author):

In the manuscript entitled 'Reconstructing a long-overlooked *Xylella fastidiosa* outbreak in Europe' Moralejo et al propose diverse sets of approaches to date the introduction in Mallorca of Xf strains that are causing Almond Leaf Scorch, a disease that has been known for a while in the USA and in Iran. The Authors estimated disease incidence in 126 almond orchards over Mallorca in 2017 and compared these data with Google Maps Street View images from previous years to find a spatio-temporal signal in disease progression. They elegantly provide a first datation of Xf in Mallorca by analyzed tree rings with a qPCR assay. The date of introduction of Xf in Mallorca (around 1993) was more or less confirmed using tip dating on genomic data. This manuscript presents an overall impressive collection of data and a very interesting angle to approach the history of Xf in Mallorca. However, in its current version this manuscript suffers a series of issues that precludes its publication:

While the manuscript is quite long, with a long supplementary material, the methodology is not sufficiently precisely described. Such highly important questions as i) how where GMSV pictures analyzed? ii) How were the 126 orchards selected? iii) what is the spatial distribution of the 34 trees for which tree rings were analyzed? iv) with which subspecies were they infected? v) is really the temporal signal significant enough to date the divergence of Mallorcan isolates from American ones? among others have no answers in the manuscript.

The Authors choose to mix the Results and the Discussion in one single section, which has the major drawback of weakening both. In most parts of this combined section, Results are not sufficiently clearly presented and interpreted. One typical example is the part dealing with fungal trunk pathogens. Finally, it seems that no analyses were made in the frame of the current study concerning these fungal pathogens, and that the Authors are only reporting and discussing previously published results. The results concerning the analysis of the tree rings are also presented in a general manner but not in a detailed way. Raw results should be provided in supplementary data to understand how the Authors ended up with the nice Figure 3. Indeed, as no indication are provided concerning the selection criteria of these trees, the interpretation of the Figure is quite difficult.

The last part of the Results and Discussion section seems to have been written by a different writer than previous parts. It is more technical but also suffers from choices that were not discussed, neither supported by references, and results were poorly discussed. For example, the Mallorcan strains first were supposed to have diverged from their Californian relatives in 2004, then constraints (the supposed date of introduction) were introduced in the models and a very similar introduction date is obtained. A date for "emergence" of Xfm in Corsica is proposed, it is highly different from one that was previously published but the divergence between these results is not discussed.

In the introduction section, the Authors indicate that they localized two orchards where Californian almond varieties were grafted onto local almond-seed rootstock in the 1990s. Then the story is focused on the Son Carrio orchard, but nothing is mentioned about the other orchard. What is its present sanitary status? This is surprising and this gap must be filled in.

Detailed comments

L. 29 I agree with the Authors that trade of infected plant material has certainly been the most probable route for introduction of Xf in Europe, however, direct evidence is still lacking for epidemics in Italy, and outbreaks in France, for example. In particular, we do not have any direct and definite evidence that infected insects were not associated with introductions in Italy or France. I suggest that the Authors rephrase this sentence.

L. 35 : please delete 'disease' as 'epidemic' refers to a disease.

L. 56: several references using appropriate and recommended methodologies present data indicating that Xf is made of 3 subspecies (Denancé et al., 2019 and Marcelletti and Scortichini,

2016). I suggest that the Authors delete the word 'main' from their sentence and refer to these papers. Ref 11 did not use ANI as recommended for taxonomic analyses, and hence their conclusion regarding this point are not appropriate.

L. 57-59: can we really said that a host range of at least 30 species is a narrow host range? This is the case of ST 53, ST7, and ST6 as known from European cases. Please rephrase this sentence as it is incorrect.

L. 124-125: "first Xf established outbreak in Europe" is not correct as Soubeyrand et al., 2018 proposed that Xf established in Corsica around 1985.

L. 151-155: the Authors should briefly mentioned how they assessed disease incidence. From supplementary materials and methods and Supplementary Data 1, it is hardly possible to understand how Authors ended with the figures reported in L. 151-155. By the way, in Supplementary Data 1, the meaning of 'N' in column D is not explained.

L. 153: 'SE' was not previously used, is it standard error of the mean (SEM) ?

L. 155-156: this sentence is not clear, if the trees were not removed, they obviously remained in the field. The Authors should clarify this sentence.

L. 158-160: the two parts of the sentence seem in contradiction. Either younger plant exhibit less mortality and the incidence is lower and this is significant, or there is no need to mention it, or something is missing in the first part of the sentence. Please clarify.

L. 168: please explain what is qGIS or QGIS (as written L. 518)?

L. 181: how was the analysis done? Please explain the methodology used to analyze the pictures? Was it done using image analysis software or by eye?

L. 152, 173, 341: these 'see below' indications are not correct. I do not understand to what these '(see below)' refer to. Please present data and information when needed. This is a consequence of having mixed 'Results' and 'Discussion' section. This strongly decreases the clarity of the demonstration.

L. 201: this sentence is unclear: Please rephrase to clarify the intent while using 'Coincidentally' at the beginning of the sentence. Most mutations in HKG are synonymous –and hence not visible by BlastN, this is expected for these genes coding proteins involved in basic cellular functions.

L. 208-212: how were selected these trees? Do these 55 isolates represent 55 trees? Are there any relationships between Cq and successful isolation?

L. 218-221: this sentence is not really clear: how were these figures (87 local and 23 foreign almond varieties) obtained? What were the thresholds that were expected?

L. 224 replace 'saplings' by 'samplings'

L. 236-247: The novel results brought on this subject in this present study are not clearly indicated. As no Material and Methods are provided for this part, it is most certainly based only on previously published data. In this regard, presenting a graph from previously published data in Supplementary Fig 4 is ambiguous, except if novel data were incorporated. The Authors should make clear if any novel results were obtained.

L. 239-241: at which scale is this diversity analyzed: per tree, per orchard or globally at the level of a region?

L. 243 why using 'Coincidentally' at the beginning of the sentence?

L. 260-262: can the Authors indicate the ST(s) obtained while typing spittelbugs?

L. 271-272: can the Authors describe a little bit this AWDI and indicate how it is supported by literature references? Can the Authors explain how they selected the time frames (1985-2003, ie 19 yrs vs. 2004-2017, ie 14yrs)?

L. 274: significantly

L. 294-295: the comparisons that were made are totally obscure to me: what are the data that were collected and what is the hypothesis that was statistically tested?

L. 296: referring to a tree picture cannot illustrate the association between ALS incidence and tree mortality. Please indicate more clearly the data you were analyzing.

L. 306-333: it is highly difficult to understand the experimental design that was used here (even when referring to SI Material and Methods) and have a precise idea of the results that were obtained. How were the 34 trees selected? What are the results that were obtained? L. 315-319: it is indicated that $12+9+1=22$ trees presented infection from 2008, 2004 and 1998, respectively.

What about the 34-22: 12 other trees that were infected and analyzed for dating infection?

L. 319-322: more details should be provided on the subspecies-specific test that was used and about the results that were obtained. When and where were each subspecies detected?

L. 356: again "coincidentally"!!! Are all the events only a succession of coincidences?

L. 365-368: how many trees were analyzed?

L. 376-469: this part is erroneously entitled Xf subsp. fastidiosa ST1. Indeed, the first sentence refers to Supp Fig. 5 that present data of several STs within Xff. It is the same for Xf subsp. multiplex ST81 (L. 431-464) that present results for various STs.

Root to tip dating suppose testing temporal signal and validating it. In Supplemental Fig. 6, $R^2=0.612$ and $R^2=0.184$ are provided for Xff and Xfm dataset, respectively, are these indicative of significant temporal signal?

In the supplementary material and Methods the Authors indicate that they use data provided for Corsica by Soubeyrand et al. 2018. Surprisingly, however, they ended with an 'emergence' date of 2000 for CFPB8417 and CFPB8418 strains from Corsica, while Soubeyrand et al. 2018 data provided an earlier date (1985) for introduction in Corsica (not specifically for these two strains, as their datasets were not genome sequences). Could the Authors be more precise in indicating the data they recover from Soubeyrand et al., 2018 study and how they explain the discrepancies in the dates obtained for Xf introduction in Corsica?

L. 482-484: again this is not correct. A previous paper, Soubeyrand et al., 2018, already mentioned an earlier date (1985) for Xf introduction in Corsica, France, which is part of Europe.

Fig 1: Quality of the pictures is not optimal, as pictures are blurred when focusing to see details of symptoms.

Fig 3: replace the commas by dots in all figures reported in this figure.

Supp Fig 1: this sentence "Core genome sequences of ST81 from isolates from almond and olive trees were identical; cross infections between wild olive and almond trees are thus suspected." should be deleted or rephrase to be more precise and in this last case the Authors should provide ample evidence showing that sharing core genome is sufficient to share host range.

Supp Fig 3A: qPCR positive (without an 's') in the Y axis legend and text. How were cells declared dead or alive?

Supp Fig 4: how were comparisons of fungal diversity among orchards made?

Supp Table 1: some cells in column (Field infections) are empty: what does this mean?

Supp table 2 legend: please indicate the reference '(ref)'. Concerning this sentence: 'Local varieties were more frequently infected than foreign ones and almond leaf scorch was less frequent in younger plantations.' Please explain how these results were obtained (which are the local and the introduced [rather than foreign] cv and what is the age for young plantations vs. older ones) and what is the p value associated to the statistical test (which one) that was run.

Supplementary Fig. 5. Legend is incomplete. Legend for part B is lacking. Bootstrap values are lac

Reviewers' comments:

Reviewer #1 (Remarks to the Author):

The authors describe the analysis of *Xylella fastidiosa* (Xf) as the causal agent of almond scorch disease (ALSD) in Mallorca, describing the potential distribution and likely entry timescale. Furthermore, the authors carried out population analysis to compare the Xf isolates identified in Mallorca to those identified in California and state that due to the close genetic proximity etc. that the Mallorca Xf was directly introduced from California in the early 1990s. They describe further details of the Xf substructure in Mallorca through MLST analysis, indicating that it represents two distinct subspecies. They also utilised Xf analysis in tree rings to provide the likely date of Xf introduction.

General comments:

Overall the study addresses an important question regarding the likely source of the Xf ALSD introduction into Mallorca, whilst considering a pathogen that is of great broad interest. However, this current study also builds on previous studies that should be more clearly highlighted to provide context. This includes a study by the authors (Landa *et al.*, 2020) that used sequence analysis of two Xf strains from Mallorca (one from almonds) that already states that Xf was likely introduced into Europe (and Mallorca) from California due to genetic proximity. Furthermore, the proposal of Soubeyrand *et al.* (2018) describing two likely introduction periods into Europe, one of which fits well with the current study should be better discussed. Overall, the manuscript is very difficult to follow in its current format and thereby the major finding is not clear. Although interesting data is presented it is difficult in the current format to assess how this contributes to that that was described previously. For instance, the main aim is not really clear to me. It seems that determining the origin may have been the main aim? But this is described in the previous publication.

Reply: The authors appreciate the comments of the reviewer. In the new manuscript version, we have modified the Introduction section to emphasise the main aims of the work. We also try to clarify the scope of this work in relation to other previously published work. Landa *et al.* (2020) paper covers the genetic relatedness of isolates of the subspecies *multiplex* from different disjunctive areas of the Mediterranean Basin and compare them with those of subspecies *multiplex* in the USA and South America. Three whole genome sequences of isolates belonging to Xf ST81 from Majorca and Menorca were included in the phylogenetic analyses. The paper concludes that all the *multiplex* introductions were independent from each other and three of them, Corsica, (Majorca + Menorca) and Alicante were genetically related with Xf isolates from California.

The scope of our manuscript is quite different and broader than that of Landa *et al.* (2020). We first reverse a misdiagnosis of an almond disease that was studied in Majorca by some of the authors between 2008 and 2010. Secondly, we provide a tangible chronology of the almond decline (caused by Xf-ST1 & ST81) and dedicate much of our efforts to demonstrate that the almond decline was indeed caused by Xf and thus the true disease previously misdiagnosed was ALSD. For that, we have gone a little bit further by introducing a new research approach to date pathogen infections in

woody plants by qPCR that had allowed to resolve the chronology of introductions of *Xf* in Majorca. We legitimately use the hallmarks of *Xf* DNA recorded in the wood to calibrate the chronology of the Majorcan isolates of *Xf* subsp. *fastidiosa* and *Xf* subsp. *multiplex* to overcome the lack of old isolates of *Xf* in the Balearic Islands. We believe that to date our approach is the most solid way to produce reliable time-relative phylogenies without a good collection of *Xf* isolates over time. Both reviewers emphasise the need to discuss the results of this paper in the context of the findings of Soubeyrand *et al.* (2018). Although we have followed with great interest this paper, the two introduction scenarios proposed have been inferred from mathematical models based on time series data covering a few years of surveillance. These proposed scenarios need to be validated with further experimental data, especially molecular phylogeny based on the whole genome sequences of several isolates.

The format, complex nature and specificity of the terminology would be extremely difficult for a general audience to follow. There are many terms that are not described, and the logic is not easy to follow. The lack of referencing in many places also makes it difficult to determine whether certain findings are from this or previous studies. In its current format it would be much more suited to a subject specific pathology-focused journal.

Reply: Our approach to demonstrate the introduction of *Xf* in Mallorca has been multidisciplinary; therefore, we agree that sometimes the terminology used could be difficult to follow. We have included some definitions in the Material and Methods and in the main text whenever it is not clear from the context. Both reviewers raise objections to the manuscript format. In the new version, we have restructured the text to improve the logical flow and clarify what the findings of this document are and which come from previously published works.

The manuscript would greatly benefit from specialist editing to improve the general English quality and flow. I have not provided any English language corrections below as the manuscript requires extensive editing. Ensuring the aims of the study and hypothesis are stated in both the abstract and throughout would help to better guide readers.

Reply: Thanks for your recommendation. We have written a new version of the manuscript to ensure the aims and hypothesis are clear enough. The manuscript has also been edited by the Springer Nature Author Editing Service before resubmission.

It would be beneficial to include all methods in the main text.

Reply: Originally, we included them as Supplementary Information to save space, but we agree that for clarity it is better to include a more detailed section of all the methods in the main text. Following reviewer recommendation, now all materials and methods appear in the main text of the manuscript.

All data should be deposited in a public repository and not available “upon reasonable request”, see detailed comments. This is essential to support reproducibility.

Reply: We agree with the reviewer. We have increased the number of Supplementary Data and described in the manuscript text where and appropriate: i) we have added a file 'Supplementary Code' with the statistics and R scripts, ii) we have deposited all data in a public repository (FigShare (<https://figshare.com/s/0b571280bcd7738a524d;10.6084/m9.figshare.12378302>)) for MLST sequence data; iii) the raw data used for figures are available both in the Supplemental Data and in FigShare (<https://figshare.com/s/a4a78390e02e8cc123d4;10.6084/m9.figshare.12488342>); iv) raw genome sequencing data used in this study has been submitted to Sequence Read Archive under the following accession numbers: SRR11931324- SRR11931339.

Specific comments:

Abstract

Line 30: "knowledge on their pathways and chronology of entry remains circumstantial". How is the pathway circumstantial when this previous study stated also that it likely was introduced from California (Landa et al., 2020)?

Reply: Thanks for your comments. This is a general statement for the diversity of *Xf* found in Europe (subsp. *fastidiosa*, *multiplex* and *pauca*), and not specifically for subsp. *multiplex*. How and when these *Xf* lineages were introduced into Europe is poorly understood and, in most cases, circumstantial. We can reasonably speculate on the origin based on the *Xf* phylogenies for some of these lineages, including those related to almonds, as did the article by Landa *et al.* (2020), or previous publications for subsp. *pauca* ST53 in Italy. However, to date there is no solid evidence on when and how most of the *Xf* detected in Europe arrived. We believe that our manuscript sheds light on one of these introductory events. Nonetheless, we have modified the text to be more precise and include the comments of the reviewer and that of the second reviewer who reasonably remarks that some introductions by insects cannot be discarded:

Page 2, lines 30-31: "The recent introductions of *Xylella fastidiosa* (*Xf*) into Europe are linked to international plant trade. However, how and when these entries occurred remain poorly understood".

Line 41: "Pierce's disease of grapevines" this is only introduced in last sentence with no context.

Reply: We appreciate this comment. Now we have rephrased the abstract:

Page 2, lines 32-34. "Here, we show how a disease affecting ~79% of almond trees in Majorca (Spain) previously attributed to fungal pathogens was indeed triggered by the introduction of *Xf* around 1993 **and the subsequent spread of *Xf* to grapevines.**"

#It is further addressed on:

Page 2, lines 36-39. "Bayesian phylogenetic inference predicted that both *Xf* subspecies found in Majorca, *fastidiosa* ST1 (95% highest posterior density, HPD: 1990-1997) and *multiplex* ST81 (95% HPD: 1991-1998), shared their most recent

common ancestors with Californian *Xf* populations related to almonds **and grapevines**".

Introduction

Line 43-44: Please add a reference.

Reply: A reference has been added (Number 1):

Lines 47-49. "Almond trees, an important icon of the agricultural landscape of Majorca, Balearic Islands (Spain), have experienced severe decline and mortality over the last 15 years¹."

Gramaje, D. *et al.* Fungal trunk pathogens associated with wood decay of almond trees on Mallorca (Spain). *Persoonia - Mol. Phylogeny Evol. Fungi* **28**, 1–13 (2012).

Line 53: Reference required.

Reply: A reference has been added (Number 7):

Olmo, D. *et al.* *Xylella fastidiosa* en las Islas Baleares. In Enfermedades causadas por la bacteria *Xylella fastidiosa* (ed. Landa BB, Marco-Noales E, M. L. M.) 231–234 (Cajamar Caja Rural, 2017).

Line 70: Reference required to the annual survey data.

Reply: We have now included a reference from a Spanish publication at the end of the paragraph **Line 72-82** (Number ³²). We did not provide it before because this is part of the official database from the LOSVIB, Balearic Government, which is updated regularly to the European Commission.

³²Juan, A., Beidas Soler, O., Olmo García, D., García López J. de D., Closa Salina, S. Baleares: adaptándose a la bacteria *Xylella fastidiosa*. *Phytoma* 317, 16-29 (2020).

Line 93: Here this should again refer to (Landa et al., 2020). Stating that "To disentangle the likely origin of *Xf* epidemics in Mallorca" gives the impression this is unknown and has not been studied previously.

Reply: Thanks for your comments. We have changed the sentence because we understand it was confusing; now it reads:

Line 97-99 "**To understand the timescale of *Xf* epidemics** in Mallorca, the spatiotemporal relationship between the disease attributed to fungal trunk pathogens and the ALSD needs to be addressed".

Here we refer to the almond decline, because there is a link between this disease and what we explained in the next phrase:

Lines 99-102 "**In this study**, we investigated whether the almond disease was actually due to the introduction of *Xf*—i.e., the spreading pathogen hypothesis— with its subsequent pathogen-induced drought by occlusion of the xylem vessels or instead caused by 'endemic' fungal pathogens activated by abnormal environmental changes"

Line 105: Is there a reference to these comments from plant health advisors or at least names of who provided this information?

Reply: We have included the name of the plant health advisor (T. Melis, *personal communication*). We have also enclosed an e-mail by him.

Line 111: It is not clear what 'almost identical' means and whether this is significant in this context.

Reply: We have changed the text for clarity:

Lines 115-118 "Two recent reports on the genomes of three isolates of *Xf* subsp. *fastidiosa* ST1, collected from Majorca, reported the 38 kb plasmid pXFAS_5235, **with the highest sequence similarities to the conjugative** plasmid pXFAS01 from isolate M23 causing ALSD in California^{39,42}"

In the referenced publication it can find more details "Plasmid synteny was confirmed by recruiting the reads against the plasmid sequences using the CLC Genomics Workbench (CLC Bio, Aarhus, Denmark), yielding a single circular plasmid sequence of 38,297 nucleotides".

#The fact that *Xf* isolates from Majorca contain a plasmid with high sequence similarity with those from isolates found on almonds in California reinforces the likely connection between *Xf* populations from both areas, but does not prove it. This is the context of the sentence.

Results & Discussion

Lines 132-133 & 264-265: These statements seem unnecessary and could be incorporated into the text to improve the flow of the text.

Reply: We have deleted it following reviewer's suggestion.

Lines 150-155: How was the mean disease incidence assessed?

Reply: In the new manuscript version, we have revised and included more details in the Material & Methods section (**page 25, lines 604-613**) We also have added a **Supplementary Code** presenting the R script and data used in the statistical analyses. Accordingly, we have modified the text as follows:

Page 8, lines 183-186: "By integrating disease symptoms previously attributed to fungal trunk pathogens as an advanced stage of ALSD (**see Material & Methods**), **the incidence for the disease in 2017 was visually determined to be approximately 79.5% ± 2.0 (mean ± SEM) (Supplementary Data 1)**

Lines 155-156: How did you estimate the number of remaining dead trees?

Reply: we apologise the sentence was meaningless. In the Material and Method section, we explain how the ALSD incidence (%) and mortality (%) in orchards were assessed (**page 25, lines 604-613**) and how we calculate the total mean mortality

(**Supplementary Code** with the R scripts and calculations). The raw data is provided in the **Supplementary Data 1**. We have modified the text for clarity.

Pages 8-9, lines 189-190: “Similarly, we extrapolated that at least ~ 552,869 dead trees remained in the fields (**Supplementary Data 1**)”

Line 170: Can the spreading centres be indicated on the Figure.

Reply: Thanks for the suggestion. We have indicated the putative spreading centres in the figure showing ALSD incidence in 2012.

Line 171: This section then jumps to tree ring analysis. Can the authors describe this analysis better i.e. the justification, hypothesis & how it was carried out? This would make it easier to follow.

Reply: We agree with this suggestion. In the new manuscript version, we have included the growth ring analysis into a new independent section entitled ‘**A combination of dendrochronology and qPCR revealed old Xf infections**’ (page 11, lines 251-276). Raw data of the growth ring analysis is provided in the new **Supplementary Data 4**. Full details of the justification, assumptions, hypothesis and how this was carried out is now provided in Material & Methods (pages 27-28, lines 647-682) and supported by a new **Supplementary Fig. 4**.

Supplementary Fig. 4. *Xylella fastidiosa* (*Xf*) DNA concentration increases centrifugally from older to younger growth rings in almond wood sections. Threshold cycles (C_q) in the qPCR² assays were strongly correlated to the age of the growth ring from which the *Xf*-DNA was extracted across 34 trees (Linear model: $F_{1,183} = 66.11$, $P = 0.0001$). Lower C_q values in qPCR indicate higher concentrations of the target DNA.

Lines 179-180: An alternative explanation could be a higher degree of susceptibility.

Reply: We also agree with the reviewer that a higher degree of susceptibility of the set of varieties in a specific location could produce different results. It is a good point and we include it as one of the untested explanations. The text now reads:

Page 10, line 217-222 “This distribution pattern could be explained considering two factors: (i) the occasional long-distance dispersal through infected grafts — a common practice in Majorca — at the beginning of the epidemic and (ii) the regional differences in the rate of expression of disease severity and mortality due to environmental and biological factors such as soil type, agricultural practices, **differing susceptibilities of almond varieties**, and precipitation”

Line 181: How was the GMSV used, it would be useful to provide the dates in the results section. Is there a study that has used this in a similar manner that could be referenced as I could imagine many issues with analysing 2D images of trees for symptoms? Is there a summary of this data that could be provided?

Reply: Thanks for your comments. We realised that the former manuscript version this was not fully explained. We have included more details in the Material & Methods section (**page 26, lines 619-632**) on how the images of Google street view in 2012 were used to estimate the disease incidence and mortality, and provided a reference of similar work where the estimation of the distribution of the pine processionary moth in France is described (Reference⁶⁰). The raw data is included now in the **Supplementary Data 2** and the R script to calculate the disease incidence (%) and mortality (%) is in **Supplementary Code**.

We have rewritten this paragraph as follows:

Page 9, lines 195-202. “Such a high ALSD incidence across the island indicated a relatively old entry of the pathogen. To estimate the rate of disease spread, we needed some historical reference to the ALSD incidence. The GSV panoramic-image repository provided the means to approximately assess the ALSD incidence and mortality in 2012, given that the pictures covered a large part of the territory that year (see **Material & Methods**). Thus, 249 orchards distributed throughout the island were **visually** examined, and the average incidence of ALSD was estimated at 53.4% ± 1.6 (mean ± SEM) for 2012 (**Supplementary Data 2**)”.

Lines 193-194: How many samples were analysed?

Reply: We have included the number of samples ($n=23$); **Page 7, line 161**.

Lines 190+ & 260-262: MLST data should be deposited in a public repository.

Reply: We normally deposit MLST data on GenBank and on the *Xylella fastidiosa*-MLST database (<https://pubmlst.org/xfastidiosa/>) when there is a new loci or MLST profile. Nevertheless, we have deposited on Figshare the MLST profiles and sequences of three samples of *Prunus dulcis* infected by ST1, ST7 and ST81, and of all *Philaenus spumarius* specimens analysed in the study that were infected by *X. fastidiosa* and harboured alleles belonging ST1 and ST81. We have deposited this information on Figshare and referred it on the XF-ACTORS Digital Research Object Portal (DROP) since data obtained in this work was partially funded by the H2020 XF-ACTORS project. The XF-ACTORS DROP is an infrastructure that aims to provide a unique entry point to search for open research data and open access documents on *Xylella fastidiosa*. The reference to the sequences was also included in the text of the manuscript (See the answer below with the modifications made on the manuscript).

Lines 249-262: This section is particularly difficult to follow.

Reply: Thank you for spotting this. We have rewritten this section in the new version: **Pages 13-14, lines 307-321**. “**ALSD is transmitted by spittlebugs**. Based on OQDS epidemiological studies in Italy²¹ and PD in Majorca⁷, *P. spumarius* was the main candidate vector for ALSD. In the transmission experiment, *P. spumarius* adults reared from nymphs and caged on ALSD symptomatic trees vectored *Xf* to healthy almond saplings. *Xf* was detected by qPCR in seven out of eight almond plants exposed to ‘infected’ *P. spumarius* and re-isolated from symptomatic leaves on six of those plants. Control ($n = 2$) and untreated ($n = 2$) almond plants were qPCR negative. *Xf* DNA was detected in all post-mortem *P. spumarius* adults ($n = 35$) used in the inoculation treatments, while all insects ($n = 10$) in two control replicates proved PCR negative. On the other hand, we investigated the prevalence of *Xf* subspecies on infected *P. spumarius* adults in the field by analysing the MLST of *Xf*-DNA within a subsample of 55 adults that were *Xf* positive for qPCR collected in or near three almond orchards. Among the positive spittle bugs analysed by MLST (25 samples), for 19 of them, some MLST loci could be amplified; two were infected by subsp. *multiplex* with alleles belonging to ST81, and 17 insect samples had alleles of subsp. *fastidiosa* ST1.”

Lines 267+: The link between ALSD and climatic conditions needs to be more clearly

described. This section is also difficult to follow and I'm not sure of the conclusion here. Furthermore, the authors state that they have developed AWDi but it is not clear what this is or how the analysis was carried out or which statistical test was performed.

Reply: Thanks for your comments. Following your suggestions and those of reviewer 2 we have modified this section. We therefore have first modified the title “**Biologically induced water stress versus climatic drought**” and we have stated more clearly the hypothesis both in the text and in the Material & Methods (**page 29-30, lines 738-749**). The point in this section is that the almond decline (attributed to the interaction of fungal trunk pathogens with drought, **Introduction Line 47-49**) indeed was a consequence of the biological-induced drought caused by the infection of *Xf*, as we hypothesised in the Introduction: **Lines 99-102** “**In this study**, we investigated whether the almond disease was actually due to the introduction of *Xf*—i.e., the spreading pathogen hypothesis— with its subsequent pathogen-induced drought by occlusion of the xylem vessels or instead caused by ‘endemic’ fungal pathogens activated by abnormal environmental changes”

On the other hand, Reviewer 2# has pointed out that the year periods are unbalanced, which we agree. We therefore have reanalysed the data, which now is provided in **Supplementary Data 5**, adjusting the periods (2003-2017) and (1988-2002).

Some effects of climatic conditions on ALSD symptom expression were mentioned in the first section of the Results “**ALSD is widespread in Majorca**” in the **page 7, lines 145-147** “An episode of high water demand (three consecutive days of $T_{max} > 33^{\circ}\text{C}$ and $\text{RH} < 60\%$; Fig. 1a) in mid-June 2017 triggered the synchronous onset of ALSD symptoms throughout Majorca”.

In the same section **lines 153-156**, we stated “ALSD symptoms emerged two weeks later in the summer of 2018 than in 2017 due to wetter and milder late-spring early-summer weather, stressing the annual variability in ALSD symptom expression and other *Xf*-related diseases driven by environmental factors^{43,44}”

We wrote again this section in the new version:

Page 15, lines 341-351: “Among farmers, the almond decline was generally attributed to the increase in drought periods due to climate change. Drought has also been partly associated with the appearance of the almond wood fungal complex⁴⁸. If there was a relationship between drought and almond decline, an increase in drought episodes would be expected beginning at the time the disease was first noticed around 2003 (see **Material & Methods; Supplementary Data 5**). To test this, we compared the crop average **cumulative water deficit index (CWDi)** during the periods **1988-2002 (before decline) and 2003-2017 (after decline)**. We found that the mean CWDi before 2003 was lower (CWDi = -263.4), indicating greater water stress, than that after 2003 (CWDi = -235.7), but this difference was not statistically significant ($t = 1.25$, $df = 28$, $P = 0.22$)”.

Lines 288-292: Where is this data?

Reply: We apologise this was not clear in the manuscript. Data was provided in the **Supplementary Data 1**. Spearman correlations is calculated from **Supplementary**

Data 1 and 2. We include the statistic tests in the **Supplementary Code**. How it is calculated is explained in Material & Methods (**page 26, lines 642-645**).

For clarity, we have modified the paragraph:

Page 10, lines 230-236: “In mid-summer 2017, we noted that trees with ALSD symptoms commonly intermingled in the same orchard with others that exhibited different stages of general decline, shoot and branch diebacks, or a combination of both (**Supplemental Data 1 & 3**). In our evaluations of disease incidence, we found a significant statistical dependence between ALSD incidence (%) and tree mortality (%) within orchards (2017 disease incidence assessment: Spearman’s rank correlation coefficient: $\rho(126) = 0.88$; $P < 0.0001$; 2012 disease incidence assessment $\rho(249) = 0.89$; $P < 0.0001$; **Supplementary Data 1 & 2**”.

Lines 292-296: Again, it is not clear where the data is that these statistics are based on? The figure panel referred to is an image rather than data.

Reply: As above, we have clarified where the raw data from which the statistical analysis is based on is included in the new version of the manuscript. We have included a section with the R script for the **Supplement Code** and more details in the Material & Methods (**page 25, lines 592-603**). We modified the sentence in the Results as follows:

Pages 10-11, lines 237-245 “Friedman’s test showed that there were significant differences among repeated measures of tree severity scores in the time series, which corresponded to the sequences of symptom development, from leaf scorch to shoot and branch diebacks to tree death, over time ($\chi = 42.41$, $df = 4$, $P < 0.0001$; Kendall's coefficient of agreement = 0.69; **Supplementary Fig. 3; Supplementary Data 3**). Furthermore, ALSD symptoms preceded shoot and branch death in 96% of cases, while trees without ALSD symptoms remained healthy until images were no longer available (**Supplementary Data 3**). Together, these results suggest a continuum in the process from systemic *Xf* infection to tree death, as illustrated in **Fig. 1c.**”

We have included a new Supplementary Fig. 3 to support the results:

Supplementary Fig. 3. Box plot showing the sequences of ALSD severity scores over time. Each point corresponds to the score of single tree ($n= 71$) and its repeated measurements in time 2, 3, 4 and 5. On the right side of the figure images representing the symptoms scale.

Lines 306+: This then jumps back to a much more detailed discussion of the tree ring dating. It would be easier to follow if this analysis was discussed in a single section.

Reply: we agree with the suggestion and made a single section (page 11-12, lines 251-276). **'A combination of dendrochronology and qPCR revealed *Xf* ancient infections,** and we have extended the explanation in the Material and Methods.

Line 318: 'Cq = 2 9' is this 2 or 29? If 2 then that would indicate a likely problem with the qPCR?

Reply: misspelling now corrected to 29, thanks (page 11, line 262).

Lines 319-320: Specific primers were utilised. Were these designed in this study or a previous study? If here then please provide a table with the sequences, if not then please provide the reference.

Reply: Thank you for spotting this. We used the duplex TaqMan qPCR assay that differentiate *Xf* isolates belonging to subsp. *fastidiosa* and *multiplex* developed by Burbank and Ortega (2018).

Burbank, L. P. & Ortega, B. C. Novel amplification targets for rapid detection and differentiation of *Xylella fastidiosa* subspecies *fastidiosa* and *multiplex* in plant and insect tissues. J. Microbiol. Methods 155, 8–18 (2018)

We have modified the text to include this reference, and extended the results (**pages 11-12, lines 264-270**). We also have added a **Supplementary Data 4D** to support those results.

“Additionally, specific primers and probes targeting genome-specific regions in a duplex qPCR assay⁴⁵ allowed the differentiation between *fastidiosa* and *multiplex* subspecies in rings of 25 trees, with nine and 19 trees showing infection by subspecies *fastidiosa* and *multiplex*, respectively, and three trees showing a mixed infection. The analysis enabled dating the infection back to 1998 (subsp. *fastidiosa* ST1, two trees) and before 2000 (subsp. *multiplex* ST81, one tree) in Maria de la Salut and Binissalem, respectively (**Supplementary Data 4D**)”.

Line 325: ‘right-censored’ is used multiple times but without a clear explanation. I am not sure what is meant by this term.

Reply: We have now described the term in a new section “**Survival analysis**” in the Material & Methods (**page 28, lines 684-690**). We have modified the text accordingly on **lines 280-283**: “Taking advantage of the *Xf* qPCR data from dated trunk rings for each felled tree, we calculated the time from infection to tree death (decrepitude). When the trees were cut down in 2017 and 2018, most were still alive, so they were right-censored in the survival analysis (**Supplementary Data 4B**).

Right censoring refers to individuals followed up from a time origin t_0 (infection) up to some later time point t_c (time trees were felled) and the individual has not had the event of interest (death/decrepitude), such that all we know is that their event has not occurred up to their censoring time t_c .

Lines 363-365: A total of 2/13 trees showed symptoms after grafting - this doesn't seem particularly high but is used as evidence that grafting from Californian trees was the cause of pathogen introduction?

Reply: We agree with the reviewer's comment that at first instance this number does not seem particularly high. We have included an explanation indicating this potential shortcome:

Page 17, lines 396-402: “In our grafting transmission experiment, buds were collected from infected trees at the beginning of June, when trees are mostly asymptomatic and the bacterial load in the vascular system is lower and thus may be irregularly distributed along the wood tissue, which could be one of the reasons for the number of successful infections being lower than expected. Nonetheless, a 15% transmission rate extrapolated to a common plantation of one hectare considerably increases the likelihood of *Xf* establishment in new areas”.

Lines 352 & 366-368: If median survival estimates are 14 years then how were trees still alive to analyse 25 years after grafting? This would be useful to discuss.

Reply: This is a good point; we did not explain it the former version because of lack of space. We have included some additional text to discuss it in the new version:

Pages 17-18, lines 402-411. “Moreover, in our further efforts to link epidemiological events between California and Majorca, we identified two orchards, one in the municipality of Consell (centre of the islands) and the other in Son Carrió, where Californian almond varieties had been previously grafted onto local rootstocks. According to a farmer in Son Carrió, the plantation began to decline several years after grafting, which led him to graft again with a local almond variety onto the Californian variety. We counted the growth rings in two wood sections of a single tree and determined 1995 ± 1 as the date of grafting. We expected to detect *Xf*-DNA in growth rings around 1995, but in one single trees analysed, the oldest ring with *Xf*-DNA was from 2006 ± 2 (**Supplementary Data 4**)”.

Lines 376-386: With all Mallorca isolates being recent how did the authors use tip dates? These would surely reflect the older US population that has a higher level of diversity? This limitation is mentioned to a degree much later in lines 451-454 but it is important to highlight within this section.

Reply: Thanks for your suggestion. Ideally, we would need enough number of isolates evenly distributed over time to obtain a good calibration for the molecular clock. However, this is not always possible. We provide the p-values in the Supplementary Fig. 8 a). We have rephrased the text as follow:

Page 18, Lines 426-430: “**Root-to-tip dating provided little temporal signal** but was significant enough to calibrate the molecular clock without tree priors (**Supplementary Fig. 8a**);). Thus, we calculated the time of the most recent common ancestor (tMRCA) for the Majorcan clade to be around 2004 (95% highest posterior density, HPD: 1982-2015)”.

Line 405: Why is the date stated (1882) outside of the confidence intervals?

Reply: We are sorry this was a mistake. We have corrected it to 95% HPD: **1878-1886**; pp=1, thanks (**page 19, line 452**)

Lines 406-410: As stated n=6 is a very small number of genetic analyses for the Californian group. The authors should also provide the number of samples for the Mallorcan clade.

Reply: Thanks for your suggestion. We have inserted the number ($n=15$) (**page 20, line 457**)

Lines 420-422: How was genetic diversity within isolates determined from the FST value?

Reply: This sentence was not precise enough; we have changed it now for:

Page 20, lines 463-477. “On the other hand, the presence of two subclades within the Majorcan clade suggests some incipient evolutionary divergence from the introductory event. **The genetic variability between and within these subclades was compared by calculating the fixation index (Fst), which measures population differences due to genetic structure.** A higher Fst (0.21) than expected was obtained. However, contrary to expectations, both subpopulations were apparently not genetically structured either by their hosts or by their geographical distribution, since the isolates of vines and almonds throughout the island were interspersed within both subclades (Fig. 5). Furthermore, when the Majorcan isolates were grouped according to the host and analysed for their genetic diversity within and between each group (vines and almond trees), we found an Fst = ~0. These data strongly suggest the existence of cross infections between almond trees and vines by ST1 strains of subspecies *fastidiosa*, a limited gene flow between both subclades and a high clonality within them, since no recombination was detected using ClonalFrameML56 (Supplementary Fig. 9a).

Lines 474-475: How do your findings reveal the invasion route? If referring to the grafting onto local rootstocks this was already hypothesised in (Landa et al., 2020) and it would be useful to understand how the current findings support this?

Reply: We regret to contradict the reviewer but Landa *et al.* (2020) did not hypothesise this. On that study, it was hypothesised that plant trade could be the origin of the introduction, but they do not mention specifically cuttings, or grafting.

In 1993 the Agriculture Department of the Balearic Islands together with almond cooperatives organised a visit to California to gain experience of the impressive almond production in California. Among people who visited California, there were grafting professionals, a technique very extended in Mallorca. It is not nice to accuse anyone without strong proves; this is not the purpose of the manuscript. However, we provide the reference of the chapter in the book where the visit of the Mallorcan voyage to California is detailed. This could be interpreted as circumstantial evidence but when you put together all the information provided in the manuscript it is hard to give a coherent alternative explanation.

Lines 482-486: Here the authors state that it is remarkable that their results support an earlier introduction of Xf, but this seems to have been hypothesised by (Soubeyrand et al. 2018) in France and it would be advantageous to mention this here.

Reply: We agree with the suggestion. We have modified this sentence at the beginning of the Conclusion remarks:

Page 23, lines 530-532 “Our research reduces the date of the first Xf outbreak in a crop in Europe by 20 years, **in line with the predictions proposed for an ancient introduction of Xf in Corsica based on mathematical models**⁵³”.

Lines 466+: The concluding remarks are very long, and it would be helpful to have a more concise summary of the major findings and their implications.

Reply: We have tried to reduce them as much as possible.

Line 532: The authors state that they didn't use all published samples, why was this? An explanation in the text would be helpful.

Reply: The reason is that we wanted to catch the more recent evolutionary history of the isolates related to ALS, i.e. the last 100 years, and not include isolates that would distort this relationship unnecessary increasing the branch length and thus the time scale. For this reason, we excluded the WGS of *Xf* subsp. *multiplex* from Toscana (Italy) as these strains are more related to isolates from the east USA.

We include an explanation in the text on: **pages 30-31, lines 773-778**. "To catch the more recent evolutionary history of the *Xf* subsp. *multiplex* isolates related to ALS, i.e., the last 100 years, without increasing the time scale, we included all genomes available at GenBank in December 2019 from Europe and isolates from California used by Landa *et al.*³¹, except those from Tuscany due to their relatedness to older east-USA *multiplex* lineages, along with four isolates from Majorca and Menorca sequenced in this work (Table 1)".

Line 558: How many is 'most' trees?

Reply: We have included a **Survival analysis** section in the Material & Methods (**page 28, lines 684-690**). The raw data with the column of censored trees is in **Supplementary Data 4B**. The exact number in the text on **page 28, lines 686-689**. "At the time the trees were felled, the event of interest, tree death (decrepitude), had not occurred in 29 of the 34 trees; therefore, they were right-censored in the survival analysis (i.e., survival was above that in 2017-2018 but by an unknown amount)".

Lines 534+: How were the genomes mapped to the reference, SNPs called etc.

Reply: All genomes were *de novo* assembled and not mapped to any reference. Fasta files obtained from the assembling process were annotated with PROKKA and protein amino-acid sequences obtained were compared with the Get homologues software using the criteria of 50% similarity over 50% of coverage alignment. Nucleotide sequences of the core genomes genes present in monocopy were further analysed in order to detect the presence of SNPs and perform genetic comparisons.

Lines 596-597: All raw sequence data must be deposited in a public repository not available "upon reasonable request". Furthermore, one supplementary file contains accessions for the assemblies used. However, whilst the assemblies are available, the WGS reads are not. These should be added to the SRA or a similar database to enable the results of the study to be verified by others as reproducible and for use in future studies.

Reply: Raw data used in this study has been submitted to Sequence Read Archive under the following accession numbers: SRR11931324- SRR11931339.

Tables & Figures

Figure 1: Add y-axis label to panel A.

Reply: y-axis was added to panel A.

Figure 2: Add distance scale bar and label to the heat map colour bar. Overall, this figure is quite unclear and additional labels could be helpful.

Reply: We have improved the Figure adding more legends, the spreading centres and a label to the heat map.

Figure 3: Adding keys would be helpful for the reader.

Reply: We have added them.

Figure 5: Having the outline of California in the background is distracting and the authors may consider removing.

Reply: We have been considering this suggestion and actually at the beginning we removed the outline of California from the figure. However, one of the Authors remarked that if we remove the outline of California we should do the same with the Majorcan and south-east USA maps. Finally we have decided to leave the Californian outline because it visually indicates that the Majorcan clade forms part of the Californian source population.

Figure 6: As above regarding the California outline.

Reply: We have removed it.

Supplementary Data 1: Why is the age of trees only given as younger or older than 30 yrs? With the distance from Son Carrio, why is this more or less than 20 km instead of the exact value?

Reply: This is a little arbitrary cut which has an agricultural and physiological explanation. First, most almond plantations in Majorca are older than 50 years. Moreover, in the 1990s there were public economic aids to replant almonds, so in the field they can be more or less easily distinguished. Second, the tendency of growth rings to decrease in width as the tree ages reaches constant growth around 25-30 years, making it easier to visually estimate if an almond plantation is under 30 years that, for example, 40 years.

We have included an explanation in the Material & Methods (page 26, lines 618-622). “Because common perception among farmers was that older trees were more affected

than younger plantations, we tested whether age had an effect on disease incidence and mortality. The orchard age was visually estimated based on the trunk diameter and classified as young (≤ 30 y) and old (> 30) plantations.”

The exact values would have made sense if there was a unique focus. We actually thought the disease began in Son Carrio but later we realized that there could be other foci. In the Material & Methods we explain it (**page 20, lines 644-646**) “We used the 20 km threshold as a reference for the approximate radius of dispersion due to transmission by vector insects in a given area as opposed to long-distance dispersal due to grafts or stowaway insects in vehicles⁶⁰.”

Supplementary Data 1: It would be useful to separate these two spreadsheets into two separate data files.

Reply: Done

Supplementary Table 2: It would be beneficial to provide the data for each variety so it can be cross-referenced with Table 1.

Reply: Table 2 is presented to reinforce the observation that most almond varieties planted in Majorca are naturally susceptible to *Xf*. Now we have indicated in the Table 1 that all varieties used in the inoculation assay were non-local.

Supplementary Figure 2: For the non-specialist it would be useful to briefly describe the differences in the legend.

Reply: As requested, we have added the description in the legend and included the name of each subspecies in the image

Supplementary Fig. 2. Colony morphotypes of *Xf* subsp. *fastidiosa* ST1 (**G-type**; white arrow) and *Xf* subsp. *multiplex* ST81 (**A-type**; black arrows) formed on Periwinkle GelRite (PWG) medium after streaking on a Petri dish the petiole extract from a single almond tree sample. The image was taken 15 days after plating showing large differences in growing rates and colony morphologies (**smooth-type: *fastidiosa* vs. pit-like type: *multiplex***) between both strains. Scale bar =1 mm.

Supplementary Figure 3: Appears to have been distorted.

Reply: We have improved the quality of it (now Supplementary Fig. 5).

Supplementary Figure 4: Do add a label to the heat map key.

Reply: label was added (now Supplementary Fig. 6).

Supplementary Figure 5: Please add a legend for panel B. How are the bootstraps illustrated here?

Reply: Sorry, we missed it. We have included Bootstraps (now Supplementary Fig. 7).

(b) ML tree for 25 genomes of *Xf* subsp. *multiplex* from (Majorca and Menorca), Corsica (France), Alicante (Spain) and California. In both phylogenetic trees the GTR +G+I was the best fit model (Likelihood Ratio Test) with 1000 bootstraps. The numbers provide the node support bootstrap values (1000 replications). Bar scales represent number of substitutions per site.

Supplementary Figure 7: The scale on the trees are missing.

Reply: We have included the scale (now Supplementary Fig. 9).

Reviewer #2 (Remarks to the Author):

In the manuscript entitled 'Reconstructing a long-overlooked *Xylella fastidiosa* outbreak in Europe' Moralejo et al propose diverse sets of approaches to date the introduction in Mallorca of Xf strains that are causing Almond Leaf Scorch, a disease that has been known for a while in the USA and in Iran. The Authors estimated disease incidence in 126 almond orchards over Mallorca in 2017 and compared these data with Google Maps Street View images from previous years to find a spatio temporal signal in disease progression. They elegantly provide a first datation of Xf in Mallorca by analyzed tree rings with a qPCR assay. The date of introduction of Xf in Mallorca (around 1993) was more or less confirmed using tip dating on genomic data. This manuscript presents an overall impressive collection of data and a very interesting angle to approach the history of Xf in Mallorca. However, in its current version this

manuscript suffers a series of issues that precludes its publication:

While the manuscript is quite long, with a long supplementary material, the methodology is not sufficiently precisely described. Such highly important questions as i) how where GMSV pictures analyzed? ii) How were the 126 orchards selected? iii) what is the spatial distribution of the 34 trees for which tree rings were analyzed? iv) with which subspecies were they infected? V) is really the temporal signal significant enough to date the divergence of Mallorcan isolates from American ones? among others have no answers in the manuscript.

Reply: We appreciate the comments and the careful review of our manuscript. We agree there is a need to include more detailed explanations on the methodology and try to answer in this new version some important questions raised by the reviewer. Please, see below a detailed description of the changes made.

The Authors choose to mix the Results and the Discussion in one single section, which has the major drawback of weakening both. In most parts of this combined section, Results are not sufficiently clearly presented and interpreted. One typical example is the part dealing with fungal trunk pathogens. Finally, it seems that no analyses were made in the frame of the current study concerning these fungal pathogens, and that the Authors are only reporting and discussing previously published results. The results concerning the analysis of the tree rings are also presented in a general manner but not in a detailed way. Raw results should be provided in supplementary data to understand how the Authors ended up with the nice Figure 3. Indeed, as no indication are provided concerning the selection criteria of these trees, the interpretation of the Figure is quite difficult.

Reply: It would likely be easier for us to separate the results from the discussion but because of the large number of evidences we provide from different scientific fields, the manuscript will enlarge too much. We believe that after editing the manuscript and including more explanations the manuscript reads better and results are more clearly presented and interpreted.

We also agree with the reviewer about the need to present the results concerning fungal pathogens better. Now we have clarified what part of the results are from a previous publication and what belongs to this manuscript. We have also provided an additional information (**Supplementary Data 4**) to show in a clearer and detailed way the results of the analysis of the tree rings.

The last part of the Results and Discussion section seems to have been written by a different writer than previous parts. It is more technical but also suffers from choices that were not discussed, neither supported by references, and results were poorly discussed. For example, the Mallorcan strains first were supposed to have diverged from their Californian relatives in 2004, then constraints (the supposed date of introduction) were introduced in the models and a very similar introduction date is obtained. A date for “emergence” of Xfm in Corsica is proposed, it is highly different from one that was previously published but the divergence between these results is not discussed.

Reply: We have substantially modified this section to discuss better our results.

In the introduction section, the Authors indicate that they localized two orchards where Californian almond varieties were grafted onto local almond-seed rootstock in the 1990s. Then the story is focused on the Son Carrio orchard, but nothing is mentioned about the other orchard. What is its present sanitary status? This is surprising and this gap must be filled in.

Reply: We agree that it would have been better to provide more information about other orchards that were grafted with Californian varieties. In 2017 most farmers were reluctant in providing information to scientific projects due to they were frightened of the eradication protocol in the implementation of the Commission Decision (EU) 2015/789, and it was not possible for us to obtain additional information from the owner of that farm and had no permit to sample it.

Detailed comments

L. 29 I agree with the Authors that trade of infected plant material has certainly been the most probable route for introduction of Xf in Europe, however, direct evidence is still lacking for epidemics in Italy, and outbreaks in France, for example. In particular, we do not have any direct and definite evidence that infected insects were not associated with introductions in Italy or France. I suggest that the Authors rephrase this sentence.

Reply: We agree with the reviewer on the comment that insects cannot be discarded as carriers of Xf with the plants being moved. Indeed, a potential cause of the introduction of ST81 from Mallorca to Menorca could have followed this pathway. To include this uncertainty, we reformulate the sentence on **page 2, lines 30-31** to “The recent introductions of *Xylella fastidiosa* (Xf) into Europe are linked to international plant trade. However, how and when these entries occurred remain poorly understood”.

L. 35: please delete ‘disease’ as ‘epidemic’ refers to a disease.

Reply: We have deleted ‘epidemic’ from the sentence

L. 56: several references using appropriate and recommended methodologies present data indicating that Xf is made of 3 subspecies (Denancé et al., 2019 and Marcelletti and Scortichini, 2016). I suggest that the Authors delete the word ‘main’ from their sentence and refer to these papers. Ref 11 did not use ANI as recommended for taxonomic analyses, and hence their conclusion regarding this point are not appropriate.

Reply: We agree with the reviewer’s suggestion, now we have added those references, and the word “main” have been deleted.

L. 57-59: can we really said that a host range of at least 30 species is a narrow host range? This is the case of ST 53, ST7, and ST6 as known from European cases. Please rephrase this sentence as it is incorrect.

Reply: We have changed the sentence to include the precise comments of the reviewer on **page 3, lines 57-60** “Xf is a genetically diverse species made up of three

subspecies¹²⁻¹⁴ and can potentially infect more than 500 plant species¹⁵. Furthermore, each subspecies is formed by multiple genetic lineages, grouped as sequence types (ST), **each with different host ranges, although most of them infect one or two known hosts**^{16,17}”

L. 124-125: “first Xf established outbreak in Europe” is not correct as Soubeyrand et al., 2018 proposed that Xf established in Corsica around 1985.

Reply: We have modified the sentence to explain better this statement on **page 6, lines 135-137**. “We show how one of the oldest established unreported Xf outbreaks in Europe could have started after infected buds of Californian varieties were grafted onto local rootstocks around 1993”

Indeed, we mention in the first manuscript that one of the Corsican introductions are likely prior to the Majorcan one in the discussion section. However, for the time being, as the reviewer mention, the work from Soubeyrand *et al.* (2018) it is a proposal based in a mathematical model which only contemplates two introduction scenarios, 1985 and 2001. These are extrapolated from a time series of surveillance data recover from 2015 to 2017. As Soubeyrand *et al.* state in their paper, these scenarios need to be validated by including whole genome sequences of a large number of isolates. Instead, our study places DNA ‘fossil’ in 1998 and trace back the origin of the two clonal lineages following the almond decline to 1993. Throughout the paper Xf is tangible. We guess that arguably we can state that this is the first demonstrated introduction of Xf in Europe based not only on genomic data but on combining different approaches including experimental work from different fields. In any case, for not contradicting or loosing merit of the paper by Soubeyrand *et al.* (2018), which is far away from our purpose, we have rewritten the sentence to adjust better to the reality.

L. 151-155: the Authors should briefly mentioned how they assessed disease incidence. From supplementary materials and methods and Supplementary Data 1, it is hardly possible to understand how Authors ended with the figures reported in L. 151-155. By the way, in Supplementary Data 1, the meaning of ‘N’ in column D is not explained.

Reply: We apologize this was not clear. We did not include details on how we assessed disease incidence in the main text for sake of brevity and flow. In the new manuscript we have included a **Supplementary Code** with the R script and reference raw data with full details on the calculation. We have also included full details on how we assessed the disease incidence in the Material & Methods (**page 25, lines 611-613**).

We have changed the headings of the **Supplementary Data 1** as follows:

Locality	Longitude	Latitude	No. trees	Incidence (%)	Tree age	Mortality (%)	Distance from Son Carrió
----------	-----------	----------	-----------	---------------	----------	---------------	--------------------------

L. 153: ‘SE’ was not previously used, is it standard error of the mean (SEM) ?

Reply: We have changed this to include the SEM (**page 8, lines 183-186**). “By integrating disease symptoms previously attributed to fungal trunk pathogens as an advanced stage of ALSD (see Material & Methods), the incidence for the disease in 2017 was **visually determined** to be approximately **79.5% ± 2.0 (mean ± SEM)** (**Supplementary Data 1**).”

L. 155-156: this sentence is not clear, if the trees were not removed, they obviously remained in the field. The Authors should clarify this sentence.

Reply: We have reformulated the sentence in the new manuscript version:

Page 8, lines 186-190. “**From this percentage**, we estimated that ~1,250,308 almond trees, including dead trees, would have been infected by *Xf* (2017 almond plantation census: 19,417 ha with an average density of 81 trees ha⁻¹; <https://www.mapama.gob.es>). **Similarly, we extrapolated that at least ~ 552,869 dead trees remained in the fields (Supplementary Data 1).**”

L. 158-160: the two parts of the sentence seem in contradiction. Either younger plant exhibit less mortality and the incidence is lower and this is significant, or there is no need to mention it, or something is missing in the first part of the sentence. Please clarify.

Reply: Thanks for the correction. The sentence has been reformulated on **page 9, lines 192-194:**

“ALSD incidence affected trees of different ages equally ($P = 0.50$); however, younger plantations (trees ≤ 30 years) suffered less mortality than plantations with trees older than 30 years ($\chi = 5.37$; $df = 1$, $P < 0.020$)”.

L. 168: please explain what is qGIS or QGIS (as written L. 518)?

Reply: We have explained it on **page 9, lines 213-215** “To further capture the spatial-temporal spread of ALSD, we mapped the estimated disease incidence (%) and mortality (%) distribution among orchards for 2012 and 2017 **using QGIS⁴⁷ geographic information system software** (Fig. 2)”. We have also added a reference for it.

L. 181: how was the analysis done? Please explain the methodology used to analyze the pictures? Was it done using image analysis software or by eye?

Reply: We have now provided more details on how this was calculated in the Material & Methods lines (**page 26, 619-632**). In the Results section, we have also rephrased it, so that the reader can follow the arguments without the need of going to the Material and Methods section.

Page 9, lines 195-202. “**Such a high ALSD incidence across the island indicated a relatively old entry of the pathogen. To estimate the rate of disease spread, we needed some historical reference to the ALSD incidence. The GSV panoramic-image repository provided the means to approximately assess the ALSD incidence and mortality in 2012, given that the pictures covered a large part of**

the territory that year (see Material & Methods). Thus, 249 orchards distributed throughout the island were **visually** examined, and the average incidence of ALSD was estimated at 53.4% ± 1.6 (mean ± SEM) for 2012 (**Supplementary Data 2**)”.

In the new manuscript version, we have moved the citation to Fig. 3. to the new section **“A combination of dendrochronology and qPCR revealed old Xf infections”**.

L. 152, 173, 341: these ‘see below’ indications are not correct. I do not understand to what these ‘(see below)’ refer to. Please present data and information when needed. This is a consequence of having mixed ‘Results’ and ‘Discussion’ section. This strongly decreases the clarity of the demonstration.

Reply: We have modified accordingly to reviewer comments.

L. 201: this sentence is unclear: Please rephrase to clarify the intent while using ‘Coincidentally’ at the beginning of the sentence. Most mutations in HKG are synonymous –and hence not visible by BlastN, this is expected for these genes coding proteins involved in basic cellular functions.

Reply: To clarify this, we have simplified the sentence on **lines 166-167, pages 7-8:** “ST81 shared an identical MLST profile with the isolate ‘Fillmore’ (Acc. n°: CP052855.1) recovered from an olive tree in California”.

L. 208-212: how were selected these trees? Do these 55 isolates represent 55 trees? Are there any relationships between Cq and successful isolation?

Reply: We have modified the sentence to clarify this (**page 8, lines 172-173**) “In total, 55 isolates (62%; $n = 89$ attempts) were obtained from different almond trees recovered from petioles of symptomatic leaves”.

There was some relationship between Cq and successful isolation. In any case we normally use plant samples showing lower Cq values (i.e. <30). We have explained this with more details in the Material & Methods (**page 25, lines 582-584**). **“We selected for the isolation leaf samples that had low Cq values in qPCR and were sampled between June and September 2017 and 2018”**.

L. 218-221: this sentence is not really clear: how were these figures (87 local and 23 foreign almond varieties) obtained? What were the thresholds that were expected?

Reply: We have modified the sentence for clarity.

Pages 12-13, lines 288-292: “To explain the high disease incidence and mortality observed, the “spreading alien pathogen” hypothesis requires Xf strains to be broadly pathogenic to more than 87 local and 23 nonlocal almond varieties that are growing throughout the island and **identified in the almond germplasm collections in Majorca**.

the aim of the almond germplasm collections is to characterise the diversity of almond varieties found in Majorca and new introduced varieties.

L. 224: replace 'saplings' by 'samplings'

Reply: this is not a mistake we meant saplings; sapling= young tree a term very common in forestry. In a search on the *Nature Communications Biology* website the term sapling provided 776 results.

L. 236-247: The novel results brought on this subject in this present study are not clearly indicated. As no Material and Methods are provided for this part, it is most certainly based only on previously published data. In this regard, presenting a graph from previously published data in Supplementary Fig 4 is ambiguous, except if novel data were incorporated. The Authors should make clear if any novel results were obtained.

Reply: We agree with comments of the reviewer. To clarify this part, we have first included in the Material & Methods a new section "**Review of fungal trunk pathogens implication in almond decline**" (page 29, Line 732-736). "We re-examined publications on the involvement of fungal trunk pathogens in almond decline in Majorca^{1-3,48} to compare them with fungal species implicated in OQDS in Italy^{49,66,67}. We also collected data from Olmo *et al.*⁴⁸ work to determine whether the regional diversity of fungal species associated with almond decline could be related to *Xf* infections".

We have included a **Supplementary Table 5** in which the fungal assemblage described in the works of Olmo *et al.* (Majorca), Nigro *et al.* (Apulia, Salento) and Carlucci (Foggia, Apulia) are listed for comparison.

We have rephrased this paragraph to clearly state that we obtained this data from assessment of previous published work. See changes on **page 14, lines 323-331**.

"**Fungal trunk pathogens are non-specifically related with almond decline.** In previous works^{2,3,48}, Koch's postulates were fulfilled for most of the pathogenic trunk fungi found associated with almond decline; however, none of them could be ascribed as the aetiological agent of the decline. Therefore, the term 'complex disease' was used in those studies. We collected data from the work of Olmo *et al.*⁴⁸ to determine whether **the regional diversity of fungal species** associated with almond decline could be related to *Xf* infection. We observed that fungal diversity increased in those areas where ALSD severity was the highest and therefore with the longest exposure time to the pathogen (**Supplementary Fig. 6; see also Fig. 2**).

L. 239-241: at which scale is this diversity analyzed: per tree, per orchard or globally at the level of a region?

Reply: We have rephrased all the paragraph **see page 14, lines 329-333**: "We collected data from the work of Olmo *et al.*⁴⁸ to determine whether **the regional diversity of fungal species associated with almond decline** could be related to *Xf* infection. We observed that **fungal diversity** increased in those areas where ALSD severity was the highest and therefore with the longest exposure time to the pathogen (**Supplementary Fig. 6; see also Fig. 2**)".

L. 243 why using 'Coincidentally' at the beginning of the sentence?

Reply: We have deleted 'Coincidentally' from the sentence.

L. 260-262: can the Authors indicate the ST(s) obtained while typing spittlebugs?

Reply: we have rephrased to include the ST(s):

Pages 13-14, lines 315-321: "On the other hand, we investigated the prevalence of *Xf* subspecies on infected *P. spumarius* adults in the field by analysing the MLST of *Xf*-DNA within a subsample of 55 adults that were *Xf* positive for qPCR collected in or near three almond orchards. Among the positive spittle bugs analysed by MLST (25 samples), for 19 of them, some MLST loci could be amplified; two were infected by subsp. *multiplex* with alleles belonging to ST81, and 17 insect samples had alleles of subsp. *fastidiosa* ST1".

L. 271-272: can the Authors describe a little bit this AWDI and indicate how it is supported by literature references? Can the Authors explain how they selected the time frames (1985-2003, ie 19 yrs vs. 2004-2017, ie 14yrs)?

Reply: We change the AWDi acronym for a new one, cumulative water deficit index (CWDi) that sounds more correct. This measures the crop water balance through a time frame (season, annually, five-years...) based on the equation of Hargreaves⁶⁸. The time frames intended to capture the pre-2003 and post-2003 almond decline (the almond decline was first noticed in Son Carrió around 2003). We agree with the reviewer suggestion that the time frames should be balanced. We therefore have reanalysed the data, which now is provided in **Supplementary Data 5**, adjusting the periods (2003-2017) and (1988-2002). The results of the statistical analysis are a little different, though the alternative hypothesis that drought affecting almond plantations was more intense after 2003 than before 2003 is rejected.

To improve the clarity of this issue and incorporate the changes in the statistical analysis we have rewritten this paragraph as follows:

Page 15, lines 343-363. "**Biologically induced water stress versus climatic drought. Among farmers, the almond decline was generally attributed to the increase in drought periods due to climate change. Drought has also been partly associated with the appearance of the almond wood fungal complex⁴⁸. If there was a relationship between drought and almond decline, an increase in drought episodes would be expected beginning at the time the disease was first noticed around 2003 (see Material & Methods; Supplementary Data 5).** To test this, we compared the crop average **cumulative water deficit index (CWDi)** during the **periods 1988-2002 (before decline) and 2003-2017 (after decline)**. We found that the mean CWDi before 2003 was lower (CWDi = -263.4), indicating greater water stress, than that after 2003 (CWDi= -235.7), **but this difference was not statistically significant (t = 1.25, df = 28, P = 0.22)**. Because it could be argued that drought periods prior to 2003 could have exceeded an irreversible threshold that activates pathogenic fungi to the present day, we looked for similar drought periods before 1988. We found that other severe episodes of water deficit anomalies occurred during 1963-

1968 and 1981-19845, without triggering any almond decline in the following decade. Surprisingly, the wettest period (i.e., highest CWDi) of the general data series between 1988 and 2017 occurred between 2004 and 2010 (see **Supplementary Data 5**), matching the period of the exponential growth of the ALSD outbreak (Fig. 3a). Regardless of the effect of weather as the main driver of ALSD epidemics, the most plausible explanation for the emergence of fungal trunk pathogens is the disturbance (i.e., induced drought) caused by infection and colonization of xylem vessels by *Xf*.

L. 274: significantly

Reply: Not necessary in the new version

L. 294-295: the comparisons that were made are totally obscure to me: what are the data that were collected and what is the hypothesis that was statistically tested?

Reply: We apologise this was not clear in the manuscript. Data were provided in the **Supplementary Data 1**. Spearman correlations is calculated from **Supplementary Data 1 and 2**. We include the statistic tests in the **Supplementary Code**. How it is calculated is explained in Material & Methods (**page 26, lines 642-645**).

For clarity, we have modified the paragraph:

Page 10, lines 231-239: "In mid-summer 2017, we noted that trees with ALSD symptoms commonly intermingled in the same orchard with others that exhibited different stages of general decline, shoot and branch diebacks, or a combination of both (**Supplemental Data 1 & 3**). In our evaluations of disease incidence, we found a **significant statistical dependence between ALSD incidence (%) and tree mortality (%) within orchards** (2017 disease incidence assessment: Spearman's rank correlation coefficient: ρ (126) = 0.88; $P < 0.0001$; 2012 disease incidence assessment ρ (249) = 0.89; $P < 0.0001$; **Supplementary Data 1 & 2**)".

L. 296: referring to a tree picture cannot illustrate the association between ALS incidence and tree mortality. Please indicate more clearly the data you were analyzing.

Reply: As above, we clarify where is the raw data from which the statistical analysis is based on. We include a section with the R script for the **Supplement Code** and more details in the Material & Methods (**page 25, lines 592-603**). We modified the sentence in the Results as follows;

Pages 9-10, lines 39-243 "Friedman's test showed that there were significant differences among repeated measures of tree severity scores in the time series, which corresponded to the sequences of symptom development, from leaf scorch to shoot and branch diebacks to tree death, over time ($\chi = 42.41$, $df = 4$, $P < 0.0001$; Kendall's coefficient of agreement = 0.69; **Supplementary Fig. 3; Supplementary Data 3**). Furthermore, ALSD symptoms preceded shoot and branch death in 96% of cases, while trees without ALSD symptoms remained healthy until images were no longer available (**Supplementary Data 3**). Together, these results suggest a continuum in the process from systemic *Xf* infection to tree death, as illustrated in Fig. 1c".

We have included a new **Supplementary Fig. 3** to support the results:

Supplementary Fig. 3. Box plot showing the sequences of ALSD severity scores over time. Each point corresponds to the severity score of a single tree ($n=71$) and its repeated measurements (**Supplementary Data 3**). On the right side of the figure images represent the symptom severity scale used (0 = healthy trees with no symptoms of ALSD; 1 = a branch or the whole canopy with leaf scorch symptoms; 2 = shoot and branch dieback affecting between 1 and 25% of the canopy; 3 = >25-75% die-back; and 4 => 75% die-back or dead trees).

L. 306-333: it is highly difficult to understand the experimental design that was used here (even when referring to SI Material and Methods) and have a precise idea of the results that were obtained. How were the 34 trees selected? What are the results that were obtained? L. 315-319: it is indicated that $12+9+1=22$ trees presented infection from 2008, 2004 and 1998, respectively. What about the 34-22: 12 other trees that were infected and analyzed for dating infection?

Reply: We agree with the reviewer that this part should be better explained as it is a pivotal part of results reported in the manuscript. In addressing this point, we have incorporated a new **Supplementary Data 4**, where raw data of the dendrochronology +qPCR analysis of the 34 trees are provided, as well as the qPCR results from two labs (LOS VIB & CSIC-IAS) in which Harper and Burbank protocols (to determine the subspecies) were conducted. We address two different questions: one is the growth ring is infected?: (yes/no) by performing Harper qPCR. If it is infected, can we know the

subspecies?: (Burbank). To calculate the frequency of infections we used the Harper Data. Secondly, we now provide full details on how the data were obtained in the Material & Methods.

In the new manuscript version, we have included the growth ring analysis into a new independent section entitled '**A combination of dendrochronology and qPCR revealed old *Xf* infections**' (page 11, lines 253-278). Raw data of the growth ring analysis is provided in the new **Supplementary Data 4**. Full details of the justification, assumptions, hypothesis and how this was carried out is now provided in Material & Methods (pages 26-27, lines 645-678) and supported by a new **Supplementary Fig. 4**.

L. 319-322: more details should be provided on the subspecies-specific test that was used and about the results that were obtained. When and where were each subspecies detected?

Reply: Thank you for spotting this. We used the duplex TaqMan qPCR assay that differentiate *Xf* isolates belonging to subsp. *fastidiosa* and *multiplex* developed by Burbank and Ortega (2018).

Burbank, L. P. & Ortega, B. C. Novel amplification targets for rapid detection and differentiation of *Xylella fastidiosa* subspecies *fastidiosa* and *multiplex* in plant and insect tissues. J. Microbiol. Methods **155**, 8–18 (2018)

#The DNA extract for the ring-growth was obtained after trees were felled in 2017 (31 trees) and 2018 (three trees). Samples were sent to the CSIC-IAS laboratory where they were analyzed using the duplex TaqMan qPCR assay to identify the subspecies when possible. We have modified the text to include this reference, and extended the results (see below). We also have added a **Supplementary Data 4** to support those results, as well as extended the description in the text (**Pages 11-12, lines 264-270**)

“Additionally, specific primers and probes targeting genome-specific regions in a duplex qPCR assay⁴⁵ allowed the differentiation between *fastidiosa* and *multiplex* subspecies in rings of 25 trees, with nine and 19 trees showing infection by subspecies *fastidiosa* and *multiplex*, respectively, and three trees showing a mixed infection. The analysis enabled dating the infection back to 1998 (subsp. *fastidiosa* ST1) and before 2000 (subsp. *multiplex* ST81) in Maria de la Salut and Binissalem, respectively (**Supplementary Data 4D**)”.

L. 356: again “coincidentally”!!! Are all the events only a succession of coincidences?

Reply: We have deleted it.

L. 365-368: how many trees were analyzed?

Reply: In the first version we recognise the meaning of the sentence could lead to confusion. This is now specified in the text:

Pages 17-18, lines 408-411: “We counted the growth rings in two wood sections of a single tree and determined 1995 ± 1 as the date of grafting. We expected to detect *Xf*-

DNA in growth rings around 1995, but in one single tree analysed, the oldest ring with Xf-DNA was from 2006 ± 2 (Supplementary Data 4).”.

L. 376-469: this part is erroneously entitled Xf subsp. *fastidiosa* ST1. Indeed, the first sentence refers to Supp Fig. 5 that present data of several STs within Xff. It is the same for Xf subsp. *multiplex* ST81 (L. 431-464) that present results for various STs. Root to tip dating suppose testing temporal signal and validating it. In Supplemental Fig. 6, $R^2=0.612$ and $R^2=0.184$ are provided for Xff and Xfm dataset, respectively, are these indicative of significant temporal signal?

Reply: We agree with the reviewer that the context is the whole subspecies. Therefore, we have now entitled the sections as ***Xf. subsp. fastidiosa*** and ***Xf subsp. multiplex***.

In the Supplementary Fig. 6 (now S. Fig.8) we had already provided in the legend the significant tests **“Supplementary Fig. 8. Testing temporal signal for molecular clock calibration using tip dating. Linear regression between the age of the samples and their root-to-tip distances for *Xf* subsp. *fastidiosa* (a) from Majorca ($n=15$) and the USA (12) ($F_{1,26}=41.08$, $P= 0.0001$) and *Xf* subsp. *multiplex* (B) from Mallorca ($n=4$), Menorca ($n=3$), Alicante (Spain) ($n=9$), Corsica ($n=3$) and the USA ($n=6$) ($F_{1,24}=5.18$, $P= 0.0324$). Root-to-tip distances were calculated in TempEst²⁰ using a Maximum Likelihood tree in PhyML²¹.**

#Although the linear relationship is significant, the few dates of sampling produce unwished wide confidence interval bars for the age estimation of the nodes in the clades. Here is why we introduced the informative tree priors to the model.

Page 18, lines 426-430: “Root-to-tip dating provided little temporal signal but was significant enough to calibrate the molecular clock without tree priors (Supplementary Fig. 8a). Thus, we calculated the time of the most recent common ancestor (tMRCA) for the Majorcan clade to be around 2004 (95% highest posterior density, HPD: 1982-2015).”

In the supplementary material and Methods the Authors indicate that they use data provided for Corsica by Soubeyrand et al. 2018. Surprisingly, however, they ended with an ‘emergence’ date of 2000 for CFPB8417 and CFPB8418 strains from Corsica, while Soubeyrand et al. 2018 data provided an earlier date (1985) for introduction in Corsica (not specifically for these two strains, as their datasets were not genome sequences). Could the Authors be more precise in indicating the data they recover from Soubeyrand et al., 2018 study and how they explain the discrepancies in the dates obtained for Xf introduction in Corsica?

Reply: In the first version, this information was provided in Table 1. Now, we have included it also in the text (**page 22, lines 514-508**): “For example, we used an intermediate introduction scenario (1993) for the Corsican clade due to the lack of references on the *Xf* strains for each of the scenarios in the mathematical model, as well as the few genomes available to infer the time of introduction⁵³”

In the Material & Methods, (**page 31, lines 798-803**): “The best-fit model was selected to include priors on node time (tree prior) adjusting the probability distribution of the node age based on data obtained in this research and the introduction scenarios for Corsica proposed by Soubeyrand *et al.*⁵³. **Because no distinction was mentioned**

between *Xf* strains in the model proposed for Corsica, we opted to include a middle introduction scenario (1993) and a normal distribution to incorporate uncertainty”.

L. 482-484: again this is not correct. A previous paper, Soubeyrand et al., 2018, already mentioned an earlier date (1985) for *Xf* introduction in Corsica, France, which is part of Europe.

Reply: We have modified this sentence at the beginning of the Conclusion remarks.

Page 23, lines 530-532 “Our research reduces the date of the first *Xf* outbreak in a crop in Europe by 20 years, in line with the predictions proposed for an ancient introduction of *Xf* in Corsica based on mathematical models⁵³”.

Fig 1: Quality of the pictures is not optimal, as pictures are blurred when focusing to see details of symptoms.

Reply: Following the reviewer's advice, we have increased the resolution of all the figures.

Fig 3: replace the comas by dots in all figures reported in this figure.

Reply: It has been changed.

Supp Fig 1: this sentence “Core genome sequences of ST81 from isolates from almond and olive trees were identical; cross infections between wild olive and almond trees are thus suspected.” should be deleted or rephrase to be more precise and in this last case the Authors should provide ample evidence showing that sharing core genome is sufficient to share host range.

Reply: Thank you for this very good observation. We do not know to what extent the accessory genome may rule the host specificity of each *Xf* genotype, so as the reviewer suggest we should be cautious. Nonetheless, in the specific case of almond-grapevines, we have been able to infect almonds with grapevine *Xf* ST1 isolates and we have also found haplotypes that infect both hosts. We have deleted the sentence in the legend since this issue is addressed in the main text.

Supp Fig 3A: qPCR positive (without an 's') in the Y axis legend and text. How were cells declared dead or alive?

Reply: we have removed 's' from qPCR positive. This figure is a conceptual model on *Xf* movement in the tree. So far the only evidence that the bacterial cells are alive in the outermost growth ring is that we can isolate the bacteria after scraping the wood of debarked stems. All trials to isolate *Xf* from the wood section fail because of microbial contamination. We have planned to include differential staining to distinguish living from dead bacterial cells in wood samples in the future.

Supp Fig 4: how were comparisons of fungal diversity among orchards made?

Reply: Fungal diversity was compared among almond production areas 'Raiguer (centre), Llevant (east) and Migjorn (south of Majorca)', as marked in the map based on the publication of Olmo *et al*⁴⁸. The purpose of the figure is to show the pie charts with the proportion of fungal species in three almond production areas according to Olmo *et al*.⁴⁸ with the map obtained in this work that shows a mixed ALS severity index in 2012. The areas with a high severity index overlaps with those with a higher fungal diversity. The strength of this relationship was not statistically tested.

Supp Table 1: some cells in column (Field infections) are empty: what does this mean?

Reply: We agree this should have been explained in the Table. We now explain it in the table as:

^b Only the scion cultivar was considered as no information of rootstock (empty cells) was available;

.....

Supp table 2 legend: please indicate the reference '(ref)'. Concerning this sentence: 'Local varieties were more frequently infected than foreign ones and almond leaf scorch was less frequent in younger plantations.' Please explain how these results were obtained (which are the local and the introduced [rather than foreign] cv and what is the age for young plantations vs. older ones) and what is the p value associated to the statistical test (which one) that was run.

Reply: removed ref (unnecessary). We also remove the sentence from the legend "Local varieties were more frequently infected than introduced ones and almond leaf scorch was less frequent in younger plantations" as well as from the main text.

.....

Supplementary Fig. 5. Legend is incomplete. Legend for part B is lacking. Bootstrap values are lacking.

Reply: now legend completed and bootstrap values added.

Reviewers' comments:

Reviewer #1 (Remarks to the Author):

The authors have largely addressed the reviewer comments. The narrative is much clearer and the paper (although still very long) is an interesting read that I'm sure will contribute greatly to the Xf introduction debate. However, it would have been useful if the manuscript was submitted in its current more polished form initially. Due to the extensive re-wording and re-formatting there are other points/comments that are much more obvious that the authors should now address.

Introduction:

Lines 69-71: When was this data generated i.e. was this over 4 years since 2016? Please add to the text.

Line 82: "almost restricted to California" doesn't seem appropriate when also found in Iran, Spain and Israel. Please re-phrase.

Results:

Line 158: Why is n=105 here and yet what seems to be the same dataset referred to on line 51 has n=119. If 105 of 119 then please state this, although this would be 88 % and not 72 % as listed.

Lines 158-160: Where is the data to support this? If published, please reference. It would also be useful to know what was found in the 28 % that were not confirmed as Xf – were these showing symptoms similar to Xf that is caused by another pathogen?

Lines 172+: Were these again all from confirmed Xf-infected plants, it was just the isolation that failed? Please clarify in the text. Where did these 89 samples come from? Were they part of the original 105?

Line 183: I think you meant "interrogating".

Lines 183+: Were all instances of ALSD symptoms always shown to be Xf? This assumption is reflected throughout the wording i.e. that ALSD symptoms are only caused by Xf. I imagine that it can be confused with fungal trunk pathogens as was the case until 2017 (and indicated later in the text) and so to extrapolate numbers based on all ALSD symptoms being caused by Xf would provide the worst case. This should be better phrased to account for the possibility that not all cases are caused by Xf. For instance, if 72% (as in the earlier section) the total number of affected trees would be much lower. This should then be considered in the rest of the section as the assumption is that all diseased trees are infected with Xf in all further calculations and also considered throughout the manuscript.

Lines 183+: It is not clear how specific the symptoms are for Xf compared to other diseases. This is an important point as the analysis is based on visual inspection.

Line 190: It is not clear how the mortality rate was determined?

Lines 195-196: It could also be due to rapid spread facilitated by environmental factors, transmission events etc.

Line 272: The wording "somehow fitting a logistic disease progression curve pattern" seems odd and imprecise. Please re-phrase.

Line 280: It would be useful to provide a clear explanation of the meaning of "right censored". There is some explanation in the methods, but it still isn't entirely clear, and an explanation should be included in the main text.

Line 284: The use of "somehow" at the start of this line seems odd. Does this then infer that the agreement is weak? Is it expressing surprise at the agreement? Please re-phrase.

Line 312: Could the authors clarify the difference between the control and untreated plants? This is slightly clarified around line 708, but the explanation could be better here.

Line 385-388: The authors name a specific expedition, although of course there could have been others that were not documented that were the source of the introduction. Maybe consider being more circumstantial in the wording here so as not to point to only one event as the likely introduction.

Line 410: How do the authors account for the ~10 year inconsistency in grafting and detection of

Xf?

Line 467: Why was this higher than expected?

Methods:

Line 621: It would be useful to have this example and reference in the main text to justify the approach.

Line 751: Were a subset of the field-collected insects tested for vectoring Xf prior to the experiment? I imagine many other issues also with using field-collected insects, can the authors discuss these limitations in the main text.

Line 768: Should be "reads" not "readings".

Figures:

Fig. 3: There are fewer time period labels than bars in the graph, so it is difficult to match with the text regarding the period of ALSD exponential growth (line 358).

Fig. 5: The maps are a distraction and confusing. For instance, the image and clade relating to Majorca sits within the image of California. It would be better if these were removed and clear labelling of the clades provided.

Fig. 6: Is Mallorca Majorca?

Supplemental figures/data:

Supplementary Fig. 1: It is not clear how the presence of Xf was "proven". Was this through qPCR? If so, was this the n=105 or n=21? (lines 158 – 161). You would then also need to add the data for the olive trees (which is currently missing from the text) or if this is from another study then a reference to this. It would be useful to support the text to also indicate which were ST1 and ST81 as you state that both were well distributed.

Supplementary Fig. 3: Please label the images with the appropriate time. What is the time measure?

Supplementary Fig. 5: I may have missed it, but I can't see where this is referred to in the text? It would be nice to see this as part of the main figures.

Supplementary Fig. 9: Unfortunately, this figure is blurred in my copy and of very poor quality. Do just check the one submitted.

Reviewer #2 (Remarks to the Author):

The Authors took into consideration and responded to all my comments. Altogether, the MS is now much easier to read and clearer. It gathers a nice piece of data.

I have very few comments at this stage:

L. 316: please replace 'on' by 'in'

L. 331-333: is not there another hypothesis? Xf stress plants, which results in cell lysis, and fungal population sizes increase thanks to increased nutrient availability? There is not necessarily a need to evoke activation of pathogenic phase, but just increased population size, the fungi being naturally pathogenic, but aggressiveness improve with population size.

Reviewers' comments:

Reviewer #1 (Remarks to the Author):

The authors have largely addressed the reviewer comments. The narrative is much clearer and the paper (although still very long) is an interesting read that I'm sure will contribute greatly to the Xf introduction debate. However, it would have been useful if the manuscript was submitted in its current more polished form initially. Due to the extensive re-wording and re-formatting there are other points/comments that are much more obvious that the authors should now address.

We appreciate the reviewer's comments and have tried to address below all the points raised.

Introduction:

Lines 69-71: When was this data generated i.e. was this over 4 years since 2016? Please add to the text.

Reply: Thanks for bringing this to our attention. Now we have added this information:

“Following the EU mandatory annual surveys (Decision EU 2015/789), more than 7,287 plant samples comprising 274 plant species have been analysed for Xf at the LOSVIB **from 2016 to 2019**”.

Line 82: “almost restricted to California” doesn't seem appropriate when also found in Iran, Spain and Israel. Please re-phrase.

Reply: We have modified the sentence for accuracy:

“**ALSD was first reported in California in the mid-1930's** ^{9,30} and more recently there have been confirmed **ALSD** outbreaks in Iran¹¹, in Alicante in eastern Spain²⁹ and in the Hula Valley, Israel³³”.

Results:

Line 158: Why is n=105 here and yet what seems to be the same dataset referred to on line 51 has n=119. If 105 of 119 then please state this, although this would be 88 % and not 72 % as listed.

Reply: We acknowledge that this was not very clear in the previous version of the manuscript. In the introduction line 51 we refer to the whole year (119 samples). In line 158 we refer exclusively to the summer samples (from 2017/06/21 to 2017/09/21).

We have change the sentence to clarify this: “**seventy-six out of 105** almond leaf samples across Majorca brought to the LOSVIB in the summer of 2017

tested positive for *Xf* in qPCR analyses (using both Harper et al.⁴⁵ and Francis et al.⁴⁶ qPCR protocols).

Lines 158-160: Where is the data to support this? If published, please reference. It would also be useful to know what was found in the 28 % that were not confirmed as *Xf* – were these showing symptoms similar to *Xf* that is caused by another pathogen?

Reply: We have not included the data because they are regularly reported from the Balearic Government to the European Commission in accomplishment of the Decision (EU) 2015/789. We believe that listing the database of samples from a public-entity do not provide much valuable information to the manuscript.

The reviewer raises very good insights and questions in this line and following ones (lines 158-190), each one related to the other, which need some clarification. We have realised that in the Material and Methods we omitted to say that the samples analysed at the Official Laboratory of the Plant Health Department of the Balearic Islands (LOSVIB) in Mallorca came from different sources i.e. viticulturists, cooperatives, gardeners, private fields, agricultural extension services, research, etc. Many of these samples were brought to LOSVIB by people not specifically trained on the ALSD. Negative samples were not investigated because, as explained, they were out of the scope of our research.

We have rephrased line 559-561 in the Material & Methods to include this information: “Plant samples from different sources, i.e. *Xf* official surveillance, cooperatives, farmers, agricultural extension services, etc., were received and processed for *Xf* analysis at the Official Laboratory of the Plant Health department of the Balearic Islands (LOSVIB).

Lines 172+: Were these again all from confirmed *Xf*-infected plants, it was just the isolation that failed? Please clarify in the text. Where did these 89 samples come from? Were they part of the original 105?

Reply: In the Material and Methods section, line 582-582 it is explained that: “We selected **for the isolation leaf samples that had low C_q values in qPCR** and were sampled between June and September 2017 and 2018.

For clarity we have modified the sentence in the main text: “In total, 55 isolates (62%; $n = 89$ attempts) were obtained from different almond trees recovered from petioles of symptomatic leaves (**qPCR positive**) **in the summers of 2017 and 2018**

We attribute most failures to isolate *Xf* to the difficulties to grow *Xf* subsp. *multiplex* ST81 in culture media (it grows much more slowly than *Xf* subsp. *fastidiosa* ST1).

Line 183: I think you meant “interrogating”.

Reply; No, we meant integrating but we realise that it likely was not clear. We have changed it for:

“by counting the trees that showed disease symptoms previously attributed to fungal trunk pathogens as an advanced stage of ALSD (see Material & Methods), the incidence for the disease in 2017 was visually determined to be approximately $79.5\% \pm 2.0$ (mean \pm SEM) (Supplementary Data 1)”.

Lines 183+: Were all instances of ALSD symptoms always shown to be *Xf*? This assumption is reflected throughout the wording i.e. that ALSD symptoms are only caused by *Xf*. I imagine that it can be confused with fungal trunk pathogens as was the case until 2017 (and indicated later in the text) and so to extrapolate numbers based on all ALSD symptoms being caused by *Xf* would provide the worst case. This should be better phrased to account for the possibility that not all cases are caused by *Xf*. For instance, if 72% (as in the earlier section) the total number of affected trees would be much lower. This should then be considered in the rest of the section as the assumption is that all diseased trees are infected with *Xf* in all further calculations and also considered throughout the manuscript.

Reply: Thank you for your insights and questions. We understand the reviewer’s reservation about our approach, and we apologize because as explained before we had omitted to specify in the Material & Method that the samples came from different sources: The LOSVIB is obligated to register and analyse all plant material that is sent to the lab, either by particulars, farmers, or for research purposes. These samples are included in a database that is regularly reported to the European Commission in accomplishment of the Decision (EU) 2015/789

It is important to notice that many samples in 2017 came from private farmers, gardeners, cooperative technicians, etc. who were not familiar with ALSD symptoms. Therefore, the 72% refers to the whole samples received at the LOSVIB, many of them collected by non-trained eyes who wanted to know whether their trees were infected by *Xf*. In contrast, in order to achieve *Xf* isolates from almond orchards across the island many samples were taken by some of the authors (EM, FA, MM) in the summer of 2017. All these samples showed characteristic leaf scorch symptoms and all them were qPCR positive, although some failures occurred in our attempts of isolation. Nonetheless, we can assume a high accuracy in our visual detection of ALS symptoms.

In the case of almond trees that show severe diebacks without leaf scorch symptoms, there could be some reasonable doubts. This is usually a little percentage of the trees within the orchards, since most trees with diebacks have attached leaves with scorch symptoms in August and September. To determine whether trees showing those symptoms in general were infected and thus assumed to be infected by *Xf*, we retrospectively monitored the symptom development over time using Google street view images. In the section “ALSD precedes symptoms caused by fungal trunk pathogens” we explained the details.

Finally, any estimate of a disease incidence using different methodology such as remote sensing, satellite and airplane images, massive qPCR samples, etc., at a regional scale incurs in some kind of assumptions. Our approach assumes that all plants showing leaf scorch on branches are infected by *Xf* (high confidence because it is very characteristic) and almost all almonds with decline, shoot and branch diebacks showed in the past leaf scorch symptoms (we demonstrated statistically this significant association). Of course, there could be almond trees that died by other causes but we believe these not differ from a basal mortality rate (<1%) that occurred before the introduction of *Xf* into Majorca.

Lines 183+: It is not clear how specific the symptoms are for *Xf* compared to other diseases. This is an important point as the analysis is based on visual inspection.

Reply: Leaf scorch is a very characteristic symptom of the ALSD that is easily identified once you get familiar with it. Other leaf diseases caused by fungi such as *Polystigmia amygdalinum* or *Stigmia carpophila* are easily identified as well. Leaf yellowing produced by mites or *Monasteria unicostata* infestation show symptoms that are quite different from ALSD. Drought causes leaf symptoms that can be confused with ALSD in fields that are poorly fertilised, but as we explain in the manuscript most drought symptoms are indeed induced by *Xf* infections. All these symptoms can be consulted in the Guide of Main Pest and Diseases of Almond Trees in the Balearic Islands (<https://www.caib.es/govern/sac/fitxa.do?codi=2807259&coduo=1155&lang=es> In Spanish and Catalan language)

Unlike other diseases and pests affecting almond trees, ALSD is a chronic disease in which every spring *Xf* colonises the new vessels formed connecting the leaf flush (see the conceptual model in the Supplementary Fig 5). The increase of the bacterial load colonizing the xylem during summer occludes water flow and affects the formation and maturation of the primary meristems for the next years. One of the main issues treated in this manuscript is the

relationship between shoot and branch diebacks attributed before to fungal trunk pathogens and *Xf*. In the previous version of the manuscript we englobed these subsections under the statement:

'The hen-and-egg dilemma' of the almond decline: who did come first, the fungal trunk pathogens or *Xylella fastidiosa*?

We think that we have provided strong proofs showing that shoot and branch diebacks are a consequence of previous infections by *Xf*, and thus the diebacks are an advance stage of the ALSD in which the action of opportunistic fungal trunk pathogens aggravates the disease.

Line 190: It is not clear how the mortality rate was determined?

Reply: We do not provide the mortality rate (the number of deaths in the population, per unit of time) but we use the % of dead trees in the orchards (data shown in Supplementary Data 1 and in supplementary R scripts) to estimate the number of dead trees remaining in the field in 2017. **This number is actually an underestimation of the trees that have died since the beginning of the outbreak, given that many dead trees were removed from the field.** We now have included this last sentence in the main text.

Lines 195-196: It could also be due to rapid spread facilitated by environmental factors, transmission events etc.

Reply: We agree with the reviewer that environmental factors can accelerate the spread of the disease, although the range of this variation is also constrained by inherent biological and ecological factors of the pathogen, host and vector(s).

We have changed the verb "indicate" for "suggest" to include uncertainty. Actually this sentence should be taken as a general statement for the introduction of the paragraph while anticipating the results in the next sections. Besides the high ALSD incidence, the limited movement of the main vector, *Philaenus spumarius*, (see the review of the EFSA PHL Panel, 2018), the deciduous nature of the host, and all the detailed information of the survival analysis, *Xf* DNA in the growth rings, observation of GSV, etc., indicated that the high incidence only could be explained under a >15-year introduction scenario.

Line 272: The wording "somehow fitting a logistic disease progression curve pattern" seems odd and imprecise. Please re-phrase.

Reply: we have changed the sentence to make it more precise.

“After plotting the infection frequencies of dated rings in the 34-almond samples, we observed that the infection frequency sharply increased from the inner older rings to the outer rings, fitting this distribution to a logistic disease progression curve from 1993 to 2017”

Line 280: It would be useful to provide a clear explanation of the meaning of “right censored”. There is some explanation in the methods, but it still isn’t entirely clear, and an explanation should be included in the main text.

Reply: We have now included it in the main text. “When the trees were cut down in 2017 and 2018, most were still alive, so the event of study (tree death) had not occurred yet, and thus were right-censored in the survival analysis (Supplementary Data 4B)”

Line 284: The use of “somehow” at the start of this line seems odd. Does this then infer that the agreement is weak? Is it expressing surprise at the agreement? Please re-phrase.

Reply: We have rephrased the sentence.

“Our results nevertheless **were closer to** those reported by Sisterson et al.⁴³ in a 6- to 7-year monitoring of almond plantations affected by ALS in California, where they found that 91% of infected trees survived to the end of the study”

Line 312: Could the authors clarify the difference between the control and untreated plants? This is slightly clarified around line 708, but the explanation could be better here.

Reply: We agree with the reviewer. For clarity we have now modified the text as:

“Both control plants exposed to uninfected insects ($n = 2$) and almond plants unexposed to insect ($n = 2$) were qPCR negative”.

Line 385-388: The authors name a specific expedition, although of course there could have been others that were not documented that were the source of the introduction. Maybe consider being more circumstantial in the wording here so as not to point to only one event as the likely introduction.

Reply: The reviewer is right saying this. We have added some words in the next sentence to be more circumstantial.

In 2017, we performed an epidemiological investigation to find connections between California and Majorca related to ALS. A relevant publication was found in which it described a visit to the Central Valley of California in August

1993 of a group of main stakeholders of the Balearic almond sector (agricultural extensionists, almond cooperative members, etc.) to learn about crop management in California ⁵¹. **Although there could have been other older non-documented visits and other pathways of introduction**, the most plausible explanation of how two coetaneous ALSD-related *Xf* strains only known at that time in California reached Majorca is that infected scions were brought from California into Majorca and grafted onto local rootstocks

Line 410: How do the authors account for the ~10 year inconsistency in grafting and detection of *Xf*?

Reply: This is a good question. We did not want to extent the explanation in the manuscript. We believe there are two likely main factors: (i) a *sampling effect*, due to lack of enough tree samples to find older infections in the growth rings, and (ii) that the new almond variety was grafted very close to the older graft leaving only a ca. 10 cm height of wood disk to the Californian variety for detecting *Xf* in the wood. *Xf* colonises irregularly the xylem tissue forming colony clusters scattered across the wood. The likelihood to find infections in growth rings older than 2006 would be expected to increase with larger volumes of wood analysed. Although the question raised by the reviewer is very interesting, we could not explore this aspect further because the almond trees in the orchard were uprooted in the fall in 2017.

Line 467: Why was this higher than expected?

Reply: Thank you for spotting this. Now we have included the explanation in the sentence:

“The genetic variability between and within these subclades was compared by calculating the fixation index (F_{st}), which measures population differences due to genetic structure. A higher F_{st} (0.21) than **expected for a founder** introduction event was obtained”.

Methods

Line 621: It would be useful to have this example and reference in the main text to justify the approach.

Reply: This suggestion has been adopted.

“To estimate the rate of disease spread, we needed some historical reference to the ALSD incidence. The GSV panoramic-image repository **has been used to assess the distribution and prevalence of other pest and diseases in a territory such the pine processionary moth in France**⁶⁹⁴⁷. **This approach** provided the means to approximately assess the ALSD incidence and mortality

in 2012, given that the pictures covered a large part of the territory that year (see Material & Methods)”.

Line 751: Were a subset of the field-collected insects tested for vectoring Xf prior to the experiment? I imagine many other issues also with using field-collected insects, can the authors discuss these limitations in the main text.

Reply: To the best of our knowledge, there is a lack of transstadial or transovarial transmission of Xf in spittlebugs (ref. 8 in the manuscript); so, the adults raised from nymphs are reasonably assumed to be free from the pathogen. Related to this, so far none of the 33 annual and biannual plant species analysed ($n=145$ samples) for Xf at LOSVIB were positive for Xf, despite *Convolvulus arvensis*, *Erodium* spp., *Portulaca oleracea*, *Heliotropium europaeum* and *Sorghum halepense* are listed as known hosts (EFSA, 2018). Finally, we have only found *P. spumarius* infected after mid-May when the adults move to feed on the almonds, which is in agreement with the behaviour of the spittlebugs associated with OQDS in Italy (ref. 21 in the manuscript).

To clarify this point, we have included the following sentence in the Material & Methods:

Page 30, line 754: “Due to the lack of of transstadial or transovarial transmission of Xf in spittlebugs⁸, we assumed that all nymphs were free from the bacterium”

EFSA (European Food Safety Authority), 2018. Scientific report on the update of the Xylella spp. host plant database. EFSA Journal 2018;16(9):5408, 87 pp. <https://doi.org/10.2903/j.efsa.2018.5408>

Line 768: Should be “reads” not “readings”.

Reply: thanks for the correction. We changed it.

Figures:

Fig. 3: There are fewer time period labels than bars in the graph, so it is difficult to match with the text regarding the period of ALSD exponential growth (line 358).

Reply: thank you for catching this. We have modified the time period labels in the figure and improved the quality of the figure.

Fig. 5: The maps are a distraction and confusing. For instance, the image and clade relating to Majorca sits within the image of California. It would be better if these were removed and clear labelling of the clades provided.

Reply: We have removed the maps and improved the labelling.

Fig. 6: Is Mallorca Majorca?

Reply: Thanks for the correction; now it is changed in the figure.

Supplemental figures/data:

Supplementary Fig. 1: It is not clear how the presence of *Xf* was “proven”. Was this through qPCR? If so, was this the n=105 or n=21? (lines 158 – 161). You would then also need to add the data for the olive trees (which is currently missing from the text) or if this is from another study then a reference to this. It would be useful to support the text to also indicate which were ST1 and ST81 as you state that both were well distributed.

Reply: we have corrected the caption to make clear that all points in the map belong to almond and wild olive trees that were qPCR positive up to 2017. We have marked in the figure the almond trees infected by *Xf* in which the subspecies was determined and added the data for the wild olive trees. All wild olive trees analysed were ST81, and all almond trees infected by *Xf* subsp. *fastidiosa* belong to ST1. The point we want to show in the figure is the overlap (mixture) in the distribution of the wild olives and the almond trees infected by both *Xf* subsp. *fastidiosa* and *Xf* subsp. *multiplex*, and the mixture of both subspecies in almonds in many areas of the island.

Supplementary Fig. 3: Please label the images with the appropriate time. What is the time measure?

Reply: Thanks for spotting this mistake. We missed to state in the figure caption that this is an ordinal scale. The attribute of interest is the rank, so there is no ratio scale. We have modified the figure caption as:

“Box plot showing the sequences of almond leaf scorch (ALSD) severity scores over time. Each point corresponds to the severity score of a single tree ($n= 71$) and its repeated measurements **in an ordinal scale** (Supplementary Data 3)”

Supplementary Fig. 5: I may have missed it, but I can't see where this is referred to in the text? It would be nice to see this as part of the main figures.

Reply: Now Supplementary Fig. 5 is referred in the text (page 11, line 259)

Thank you for the recommendation. We prefer to include the figure in the Supplementary information because we are trying to develop further the conceptual model in a research that is currently undergoing.

Supplementary Fig. 9: Unfortunately, this figure is blurred in my copy and of very poor quality. Do just check the one submitted.

We checked the quality of Fig. 9 and it has been improved.

Reviewer #2 (Remarks to the Author):

The Authors took into consideration and responded to all my comments. Altogether, the MS is now much easier to read and clearer. It gathers a nice piece of data.

I have very few comments at this stage:

L. 316: please replace 'on' by 'in'

Reply: it was changed, thanks!

L. 331-333: is not there another hypothesis? Xf stress plants, which results in cell lysis, and fungal population sizes increase thanks to increased nutrient availability? There is not necessarily a need to evoke activation of pathogenic phase, but just increased population size, the fungi being naturally pathogenic, but aggressiveness improve with population size.

Reply: Thanks for the comment. We realise that we have used fungal diversity and we should have been a little more precise “diversity of fungal trunk pathogens”. We have now included this change in the text.

#Most of these fungal trunk pathogens have an endophyte stage, i.e. they remain as quiescent infections in the bark or in the wood (not so clear whether in the xylem dead vessels or in the parenchyma or ray cells). Knowledge on

their ecology and pathology has increased recently (Ref. 4 in the manuscript). There is scientific consensus that most fungal trunk pathogens are opportunistic pathogens that colonise stressed plants, being water stress one of main environmental drivers (Ref. 4 in the manuscript). More than an increase of the population size, which of course occurs as a consequence of an increase of nutrient availability, in this sentence we focus on the physiological process that triggers the change from a quiescent infection to an extensive colonisation. We use the term 'pathogenic stage' in a metaphoric way to differentiate it to the endophytic commensal stage. The important question is what triggers the weakening in the plant basal defence system allowing the fungal colonisation. Our main hypothesis is that induced water stress due to *Xf* infection is what triggers the colonisation of the fungal trunk pathogens year after year. That is why we dedicate a section "Biologically induced water stress versus climatic drought" to put in context this important issue.

We appreciate the time and effort of the editor and the reviewers and for their careful reading and valuable insights and comments of our manuscript. We believe that our manuscript has improved after this and thorough and deep review.

REVIEWERS' COMMENTS:

Reviewer #1 (Remarks to the Author):

The authors have carefully addressed all points raised. I very much look forward to seeing the manuscript published in due course.